# A comprehensive study of SARS-CoV-2 main protease (M^pro) inhibitor-resistant mutants selected in a VSV-based system

**Francesco Costacurta[1], Andrea Dodaro[2]ᵒ, David Bante[1]ᵒ, Helge Schöppe[3]ᵒ, Ju-Yi Peng[4]ᵒ, Bernhard Sprenger[5]ᵒ, Xi He[4], Seyed Arad Moghadasi[6], Lisa Maria Egger[7], Jakob Fleischmann[8,9], Matteo Pavan[2], Davide Bassani[2], Silvia Menin[2], Stefanie Rauch[1], Laura Krismer[1], Anna Sauerwein[1], Anne Heberle[1], Toni Rabensteiner[1], Joses Ho[10], Reuben S. Harris[11,12], Eduard Stefan[8,9], Rainer Schneider[5], Theresia Dunzendorfer-Matt[7], Andreas Naschberger[13], Dai Wang[4], Teresa Kaserer[3], Stefano Moro[2], Dorothee von Laer[1], Emmanuel Heilmann**[1,13]*

**1** Institute of Virology, Medical University of Innsbruck, Innsbruck, Tyrol, Austria, **2** Molecular Modeling Section (MMS), Department of Pharmaceutical and Pharmacological Sciences, University of Padua, Padova, Italy, **3** Institute of Pharmacy/Pharmaceutical Chemistry and Center for Molecular Biosciences Innsbruck (CMBI), University of Innsbruck, Innsbruck, Tyrol, Austria, **4** Department of Infectious Diseases and Vaccines Research, MRL, Merck & Co., Inc., Rahway, New Jersey, United States of America, **5** Institute of Biochemistry and Center for Molecular Biosciences Innsbruck (CMBI), University of Innsbruck, Innsbruck, Austria, **6** Department of Biochemistry, Molecular Biology and Biophysics, Institute for Molecular Virology, University of Minnesota, Minneapolis, Minnesota, United States of America, **7** Institute of Molecular Biochemistry, Biocentre, Medical University of Innsbruck, Innsbruck, Austria, **8** Institute of Molecular Biology, University of Innsbruck, Innsbruck, Tyrol, Austria, **9** Tyrolean Cancer Research Institute (TKFI), Innsbruck, Tyrol, Austria, **10** Bioinformatics Institute, Agency for Science Technology and Research, Singapore, Singapore, **11** Department of Biochemistry and Structural Biology, University of Texas Health San Antonio, San Antonio, Texas, United States of America, **12** Howard Hughes Medical Institute, University of Texas Health San Antonio, San Antonio, Texas, United States of America, **13** Division of Biological and Environmental Sciences and Engineering (BESE), King Abdullah University of Science and Technology (KAUST), Thuwal, Saudi Arabia

ᵒ These authors contributed equally to this work.
* emmanuel.heilmann@i-med.ac.at

**Data Availability Statement:** The authors confirm that all data underlying the findings are fully available without restriction. All relevant data are

## Abstract

Nirmatrelvir was the first protease inhibitor specifically developed against the SARS-CoV-2 main protease (3CL^pro/M^pro) and licensed for clinical use. As SARS-CoV-2 continues to spread, variants resistant to nirmatrelvir and other currently available treatments are likely to arise. This study aimed to identify and characterize mutations that confer resistance to nirmatrelvir. To safely generate M^pro resistance mutations, we passaged a previously developed, chimeric vesicular stomatitis virus (VSV-M^pro) with increasing, yet suboptimal concentrations of nirmatrelvir. Using Wuhan-1 and Omicron M^pro variants, we selected a large set of mutants. Some mutations are frequently present in GISAID, suggesting their relevance in SARS-CoV-2. The resistance phenotype of a subset of mutations was characterized against clinically available protease inhibitors (nirmatrelvir and ensitrelvir) with cell-based, biochemical and SARS-CoV-2 replicon assays. Moreover, we showed the putative molecular mechanism of resistance based on in silico molecular modelling. These findings have implications on the development of future generation M^pro inhibitors, will help to

within the paper, its Supporting Information files or provided in a public repository. Specifically, TTMD data can be found on Zenodo: All data: https://zenodo.org/doi/10.5281/zenodo.8205370 Nirmatrelvir data: https://zenodo.org/records/8205371 Ensitrelvir data: https://zenodo.org/records/12527121.

**Funding:** This work was funded by the Austrian Science Fund (FWF) grants P35148 (D.v.L.), P34376 (T.K.) and I5406 (T.D-M.), with DOIs 10.55776/P35148, 10.55776/P34376 and 10.55776/I5406. For open access purposes, the author has applied a CC BY public copyright license to any author accepted manuscript version arising from this submission. The work was furthermore funded by the National Institute of Allergy and Infectious Disease grant U19-AI171954 (R.S-H.). The salaries of E.H. and F.C. were partly and fully covered by P35148, respectively. J.F. and B.S. were paid by FWF grant I5406. H.S. was paid with FWF grant P34376. The salary of S.A.M. was covered by grant U19-AI171954. J.P. is supported by the Postdoctoral Program of Merck Sharp & Dohme LLC, a subsidiary of Merck & Co., Inc., Rahway, NJ, USA. The funders had no role in study design, data collection and analysis, decision to publish, or preparation of the manuscript.

**Competing interests:** D.v.L. is founder of ViraTherapeutics GmbH. D.v.L serves as a scientific advisor to Boehringer Ingelheim and Pharma KG. E.H. and d.v.L have received an Austrian Science Fund (FWF) grant in the special call "SARS urgent funding". E.H. is a registered consultant at Guidepoint. S.A.M and R.S.H. are inventors on the patent application "Live cell assay for protease inhibition", application number WO/2022/094463. J.P., X.H. and D.W. are employees of Merck Sharp & Dohme LLC, a subsidiary of Merck & Co., Inc., Rahway, NJ, USA. X.H. and D.W. hold stocks of Merck & Co., Inc., Rahway, NJ, USA. X.H. and D.W. are inventors on the patent application "Coronavirus replicons for antiviral screening and testing". This work is not in any way sponsored or to any extent influenced by any of the authors' employment statuses. All other authors declare they have no competing interests.

understand SARS-CoV-2 protease inhibitor resistance mechanisms and show the relevance of specific mutations, thereby informing treatment decisions.

## Author summary

Studying dangerous viruses comes with risks and strict safety requirements. This is also true when new medications against viruses are developed and their effectiveness over time has to be tested. Unfortunately, viruses are quick to mutate and become resistant against almost any new medication. Ideally, this information is available before the medication is widely available. Then, when resistant viruses arise, patient caretakers can switch to other available treatments. However, to study the development of resistance with dangerous viruses is considered as 'gain-of-function' research, which is highly controversial and experiments have to be performed with high biological containment to prevent biosafety breaches. To facilitate studying the development of resistance and circumvent gain-of-function research of dangerous viruses, we describe a new, safe method. With this method, we can generate resistance data without using the actual virus, namely SARS-CoV-2 (alias 'Corona'). We provide resistance data against two clinically used antiviral medications that are relevant for the treatment of SARS-CoV-2.

## Introduction

Severe acute respiratory syndrome coronavirus 2 (SARS-CoV-2), the causative agent of COVID-19 (coronavirus disease-19), has established itself as a permanent human and animal pathogen worldwide. Vaccines, alongside monoclonal antibodies, have drastically reduced hospitalization and/or mortality, especially in immunocompromised individuals, the elderly, and people with pre-existing medical conditions [1]. As vaccines do not confer complete immunity against infection, the virus continues to spread effectively due to easier host-to-host transmission [2], and immune-escaping variants, such as Beta, Delta and Omicron [3–5].

In December 2021, the FDA granted emergency use authorization to Paxlovid, an orally administered medication that combines nirmatrelvir [6], the active component, and ritonavir, a pharmacokinetic enhancer. Nirmatrelvir is a highly potent protease inhibitor (PI) against the SARS-CoV-2 3CL^pro/M^pro, and is most relevant in the clinical setting, as it has been shown to decrease hospitalization, in-hospital disease progression and death [7]. One year after the licensing of Paxlovid, another protease inhibitor, ensitrelvir, was approved in Japan through the emergency regulatory approval system under the commercial name Xocova [8], and recently received fast track designation by the FDA. Recently, leritrelvir [9] and simnotrelvir [10] have been approved by the China National Medical Products Administration. Other oral M^pro inhibitors that have entered clinical trials include bofutrelvir (FB2001) [11], pomotrelvir (PBI-0451 [12]), clinical development suspended [13]), EDP-235 [14], and HS-10517 (GDDI-4405). SARS-CoV-2 continues to spread, and it is expected that the use of protease inhibitors could eventually lead to the selection of drug-resistant M^pro variants, with serious consequences for individuals who cannot benefit from vaccines due to immune defects or are at higher risk due to pre-existing comorbidities.

To date, several studies have addressed the issue of nirmatrelvir-resistant/escaping variants using different approaches: highlighting mutational hotspots [15,16], *in silico* investigation of specific mutations [17], studying resistance phenotypes [18–21], addressing which mutations

are the most prone to decrease inhibitor susceptibility [22–24] or more comprehensive studies describing either mutation resistance, fitness costs, or both [25–28]. Notably, a few groups have performed gain-of-function selection experiments using the Wuhan-1 (wild type, WT) strain of SARS-CoV-2. While this represents the most straightforward system to generate and study drug-resistant mutants, it requires government approval, BLS-3 facilities, and demands absolute caution to avoid biosafety breaches and potential release of these protease inhibitor-resistant variants.

Furthermore, owing to safety concerns, most of the studies that employed live SARS-CoV-2 were not performed using the Omicron variant [18,19,21]. In a preprint by Lan et al.[29], the authors validated the antiviral resistance of certain M^pro resistant variants in the Omicron-M^pro context by using SARS-CoV-2 replicons, circumventing potential safety issues for mutated transmissible variants. It is worth noting that the Omicron signature mutation (P132H) does not confer resistance to common inhibitors [30–33] but alters the thermal stability of the protease, as reported by Sacco et al. [34]. This alteration, as well as other structural and biochemical features caused by the P132H substitution, could potentially affect the mechanism of resistance development and the relevance of mutations selected with Wuhan-1 M^pro for Omicron.

Recently, we described a BSL-2 mutation selection system based on a M^pro-dependent chimeric vesicular stomatitis virus (VSV-3CL^pro/VSV-M^pro) [35]. Replication of the chimeric VSV was effectively inhibited by nirmatrelvir. Using this system, we were able to select a panel of mutants and characterize them through computational and cellular methods. In the present study, we deepened our understanding of the SARS-CoV-2 M^pro resistance mutation landscape by selecting mutations in the Omicron M^pro and increasing the resistance phenotypes of an already resistant variant, namely VSV-L167F-M^pro. By generating and characterizing the mechanism of Omicron-M^pro and L167F-derived mutations, individually or in combination, this study aims to advance the understanding of protease-inhibitor-resistant M^pro variants and aid the development of next-generation M^pro inhibitors.

## Results

### A safe method to select protease inhibitor resistant M^pro variants

In a previous study, we developed a safe alternative to using un-attenuated SARS-CoV-2 for the selection of inhibitor-resistant variants [18,19,35]. The technology is based on a chimeric VSV-M^pro that encodes an artificial, non-functional polyprotein (G-M^pro-L), and relies on M^pro activity for its replication (**Fig 1A and 1B**). In the absence of an inhibitor, the protease processes the polyprotein, releasing G and L, and the virus can replicate. However, when a protease inhibitor is applied, the protease is inhibited, and the virus can no longer replicate.

### Wuhan-1 and Omicron VSV-M^pro variants are equally susceptible to nirmatrelvir

To gain a deeper understanding of the potential of the Omicron-M^pro to acquire resistance mutations, we introduced the Omicron-M^pro signature mutation P132H into chimeric VSV-M^pro, generating VSV-Omicron-M^pro (VSV-O-M^pro) for subsequent selection experiments with the protease inhibitor nirmatrelvir (**S1A Fig**). First, nirmatrelvir dose response studies were performed with the Omicron and Wuhan-1 (WT) VSV-M^pro in the presence of nirmatrelvir and it was found to maintain its efficacy against both, as described previously using WT SARS-CoV-2 [30–33] (**S1B and S1C Fig**).

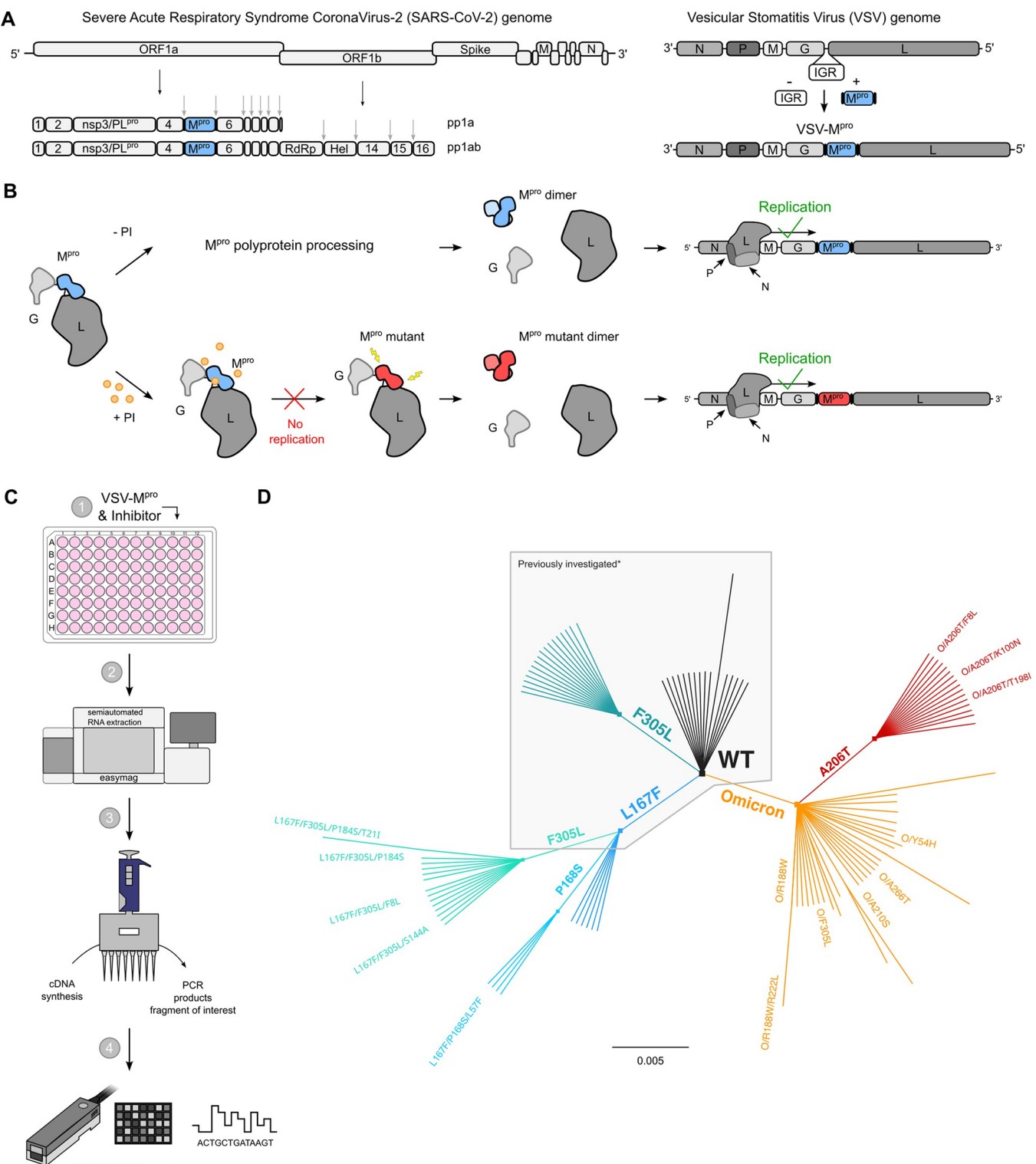

**Fig 1. VSV-G-M^pro-L construct: molecular mechanism, sequencing workflow, and mutant lineage phylogeny.** (**A**) Schematic representation of SARS-CoV-2 genome, polyproteins pp1a and pp1ab and the VSV-based mutation selection tool (VSV-M^pro). The InterGenic Region (IGR) between the genes G and L of WT VSV was replaced with the SARS-CoV-2 M^pro Wuhan-1 (also referred to as WT, wild type) sequence and its cognate autocleavage sites. M^pro genome positions in SARS-CoV-2 and in VSV-M^pro are highlighted in light blue. (**B**) The virus is fully dependent on M^pro for replication. Upon translation of G-M^pro-L, two outcomes are possible: 1. without an inhibitor, M^pro is free to process the polyprotein, and the transcription and replication complexes can assemble; 2. with an inhibitor, M^pro is inhibited, the polyprotein is not processed, and the virus is thus not able to replicate, unless it acquires a mutation rendering the M^pro

less susceptible to the inhibitor. Then, the virus can replicate despite the inhibitor. (**C**) Selection experiments workflow: BHK21 cells are infected with VSV-M^pro and treated with a protease inhibitor (nirmatrelvir), supernatants of cytopathic effect (CPE) positive wells are used to isolate viral RNA, synthesize cDNA and PCR amplify the region of interest (M^pro) that will be sequenced using Nanopore sequencing. 96-well plate was modified from public domain artwork at https://commons.m.wikimedia.org/wiki/File:96-Well_plate.svg. (**D**) Unrooted phylogenetic tree showing the relationship between original/parental viruses and mutants. M^pro variants belonging to the same parental virus are colored accordingly: WT (black), F305L (sea green), L167F (blue), L167F/F305L (light sea green), L167F/P168S (light blue), Omicron (orange), Omicron/A206T (red). For clarity, only the names of the mutants investigated in this work are displayed. *Previously generated/investigated set of mutants in our first study [35].

### VSV-M^pro supports the main protease evolution of Omicron variant and variants with existing resistance to escape nirmatrelvir

Following up on our first study [35], a previously identified resistant variant of VSV-M^pro (VSV-L167F-M^pro) was used. This mutation was of particular interest because it was also found in resistance studies with authentic SARS-CoV-2 [18,19,21]. Then, selection experiments using both VSV-O-M^pro and VSV-L167F-M^pro were performed (**S2A Fig**). By passaging both VSV-L167F-M^pro and VSV-O-M^pro in the presence of suboptimal concentrations of nirmatrelvir, M^pro mutations that were generated by the error-prone VSV polymerase were selected [36,37]. Samples from the first two pilot experiments with VSV-L167F-M^pro and VSV-O-M^pro were sequenced via Sanger sequencing. Samples from subsequent selection experiments were deep-sequenced (Nanopore) (**Fig 1C**) on the same target region ($G_{Cterm}$-M^pro-$L_{Nterm}$) (**S1 Table**). We obtained 29 distinct non-synonymous mutations in double-, triple-, and quadruple mutated VSV-L167F-M^pro variants and 47 distinct non-synonymous single- and double-mutated VSV-O-M^pro variants, as schematically shown by the unrooted phylogenetic tree (**Fig 1D**).

To achieve multiple-mutated viruses, several interesting mutants were selected according to specific parameters for further selection experiments, where we increased the concentration of the inhibitor, imposing stronger selection pressure at each passage. An overview on the M^pro-genome location of the generated mutations is displayed in **S2B Fig**. The criteria used for selecting interesting mutants were the following. First, the proximity of the amino acid substitution to the inhibitor (within 5 Å of the catalytic site or near the catalytic site, within 5–10 Å) (**Fig 2A**). Second, the prevalence in the GISAID database of specific mutations [38–40]. We considered substitutions whose count in the GISAID database was above 500 entries. Third, the frequency of a specific mutation occurring in different samples independently.

Some mutations combined more than one criterion. For example, some substitutions located in the catalytic site were also frequent in the GISAID repository. However, residues located in this area were generally more conserved compared to other coronavirus main proteases (**S3 Fig**) [41,42], whereas residues outside the catalytic site were more frequent in our selection experiments and more frequent in GISAID. We found ten substitutions with more than 500 entries: T21I, P184S, K100N, T198I, F8L, A234T, A194S, A210S, A266T, and P168S (**Fig 2B**) and broke them down according to their appearance in previous variants of concern (**S2 Table**). Despite not reaching 500 entries in GISAID, we also included the F305L mutation for further study. We recently described F305L as an autocleavage site optimization mutant with similarities to T304I selected in authentic SARS-CoV-2 resistance studies [18].

Based on the abovementioned parameters (proximity, GISAID frequency and/or independent occurrences), we chose two VSV-L167F-M^pro variants, L167F/F305L and L167F/P168S for further selection experiments. From VSV-O-M^pro, the variants O/A206T, O/R188W, O/Y54H, O/A210S, and O/A266T were chosen for further study. Among these variants, we selected O/A206T for additional selection experiments, as it occurred 17 times independently in VSV-O-M^pro selection experiments (**Fig 2C**). To confirm that this variant was not present in our VSV-Omicron-M^pro stocks already, Nanopore sequencing was performed on them. We confirmed the absence of any VSV-O-M^pro variant subpopulation to the extent of sensitivity

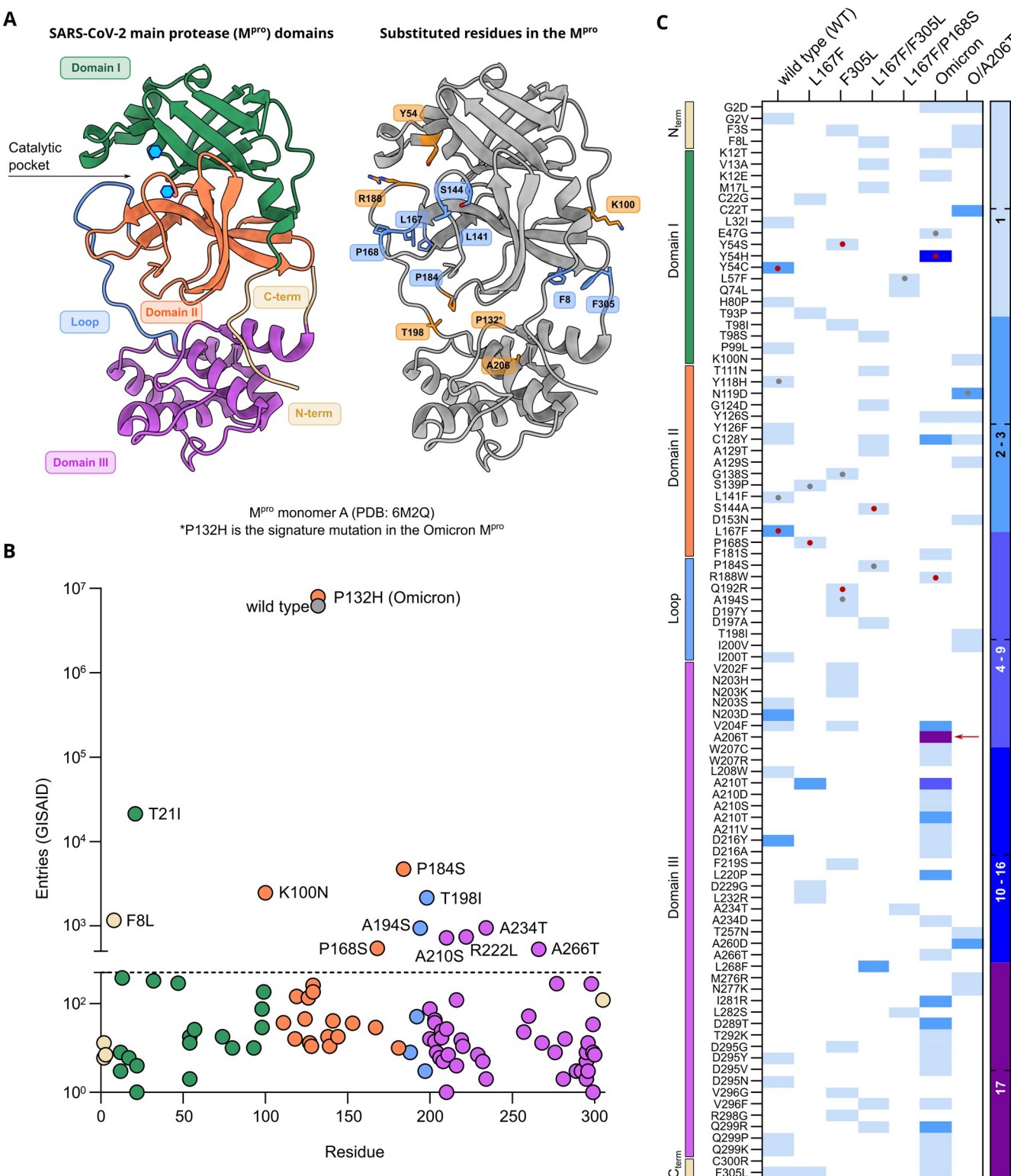

**Fig 2. M^pro structure, Frequency of M^pro mutations in the GISAID database and generated mutations.** (**A**) 3D ribbon structure of the M^pro monomer A (PDB: 6M2Q [44]). On the left side of the panel, M^pro is colored according to described domains [42]. Catalytic dyad (H41 and C145) is indicated by the light blue hexagons. On the right side of the panel, the same M^pro structure is shown highlighting some of the residue positions that were found to be mutated after selection experiments: residues found mutated in the WT background are colored blue; mutants in the Omicron background are colored in orange. Mutants at these residues were investigated either alone or in specific combinations. (**B**) The total number of nsp5/M^pro GISAID substitution entries for each

mutation found after selection experiments. Entries are colored according to domains as displayed in panel **A**. WT and Omicron-M^pro sequences are displayed in grey and orange, respectively. The dotted line at 500 represent the cut-off value. (**C**) Heat map representing the number of a specific substitution's independent occurrences from different samples of VSV-M^pro selection experiments. Wild type and F305L columns comprise mutants that were selected in previous work [35], as also shown in **Fig 1D**. Color-coded domains are indicated as in **A**. The red arrow indicates the substitution A206T. Red dots indicate residues within the catalytic site and grey dots indicate residues near the catalytic site.

that Nanopore sequencing provides. Subsequently, the next round of selection experiments with the variants L167F/F305L and L167F/P168S were carried out, in which we selected the triple-mutant variants L167F/F305L/F8L, L167F/F305L/S144A, L167F/F305L/P184S, and L167F/P168S/L57F. From VSV-O/A206T-M^pro, the mutants O/A206T/F8L, O/A206T/K100N and O/A206T/T198I were selected. All mutations occurred during selection experiments, except for T21I and R222L. R222L arose during plaque purification of VSV-Omicron-R188W-M^pro. T21I appeared during the generation of VSV-L167F/F305L/P184S-M^pro from its plasmid (also referred to as "rescue" [43]). This triple-mutated virus variant had to be produced from its plasmid, as it could not be plaque purified.

To assess whether mutants generated from the selection experiments and selected for further analyses were not detrimental for viral propagation, the Ultrafast Sample placement on Existing tRee (UShER) tool was applied. Using UShER, we generated pedigrees of the most frequently represented mutants in the GISAID database, T21I, T198I, P184S, K100N, and F8L (**S4A and S4B Fig**). The prevalence of these mutations before and after the Omicron surge (21^st December 2021, up to 18th January 2023) (**S4C Fig**) was also examined. From this analysis we observed that M^pro variants bearing these substitutions allow for continued transmission in patients, suggesting that they do not impede spread of the virus.

## Mutations selected in VSV-G-M^pro-L confer resistance to nirmatrelvir & ensitrelvir

To quantify the resistance of the selected variants with VSV-M^pro (**S5A Fig**), the most interesting mutations were introduced into two previously described live cell-based protease activity assays, namely 3CL/M^pro-On and 3CL/M^pro-Off [45] (**S5B and S5C Fig**). These two protease activity measurement tools rely on replication-incompetent VSV-dsRed variants missing either the phosphoprotein (P) or the polymerase (L), which were replaced with the red fluorescent protein dsRed. Briefly, cells were transfected either with a plasmid encoding the P protein modified with an INTRAmolecular-M^pro-tag (P_{Nterm}:M^pro:P_{Cterm}) or an INTERmolecular fusion protein made of the green fluorescent protein, M^pro and L (GFP-M^pro-L). The cells were then infected with VSV-ΔP or VSV-ΔL, respectively. The P_{Nterm}:M^pro:P_{Cterm} intramolecular tag in combination with VSV-ΔP-dsRed constitutes a gain-of-signal assay also called "M^pro-On", whereas the artificial polyprotein (or fusion protein) GFP-M^pro-L in combination with VSV-ΔL-dsRed constitutes a loss-of-signal assay also called "M^pro-Off".

First, the M^pro-On system was used to quantify the resistance phenotype of four triple mutants that arose from the WT protease: L167F/F305L/F8L, L167F/P168S/L57F, L167F/F305L/P184S, and L167F/F305L/S144A, against nirmatrelvir and ensitrelvir.

We also included the related single and double mutants to investigate the contribution of each mutation to the resistance phenotype. Single substitutions conferred low resistance to protease inhibitors, but the resistance increased by combining them. Mutations that arose during selection experiments with the Omicron protease (P132H) were also first characterized with this gain-of-signal system. In **S6 Fig**, non-linear regression analyses of M^pro-On dose-response experiments are shown. However, there were limitations in quantifying the resistance of some WT-related combinations using the gain-of-signal assay. This was largely observed in

multiple mutant variants, where the gain in signal was completely suppressed by the high resistance of these protease variants and IC$_{50}$ values could not be quantified. Since inhibitor concentrations of 100 μM or higher were cytotoxic, thereby interfering with signal generation, we did not increase the inhibitor concentration further.

Instead, to attain better resistance quantification, we cloned and tested the same set of WT M$^{pro}$ mutations in the more sensitive loss-of-signal assay (S5B Fig). As expected, while single mutants had a mild effect on M$^{pro}$ susceptibility to nirmatrelvir and ensitrelvir, triple mutants showed increased resistance compared to single mutants (S7 Fig). Initially, the most prominent differences between protease inhibitors were observed for L167F/F305L/F8L with 9.2-fold and 30.6-fold, and L167F/P168S/L57F, with 18.1-fold changes and 23.0-fold changes for nirmatrelvir and ensitrelvir (Figs 3A and S7A–S7D), respectively. When T21I was added to L167F/F305L/P184S, the resistance phenotype substantially increased, with 47.5- and 38.4-fold changes in the IC$_{50}$ values of nirmatrelvir and ensitrelvir, respectively (Figs 3A and S8A). O/A206T, O/A206T/K100N, O/A206T/T198I, O/A206T/F8L and O/Y54H/F305L showed IC$_{50}$ fold-changes of 25.9, 21.3, 46.2, 123.7 and 8.7 against nirmatrelvir, respectively, and substantially lower fold-changes against ensitrelvir. Moreover, we observed an opposite difference between responses against nirmatrelvir and ensitrelvir, where the variant O/Y54H/F305L was more resistant against ensitrelvir with an IC$_{50}$ fold-change of 15.0 (Figs 3B, S7E–S7H and S8B).

## Protease inhibitor resistance mutations can alter M$^{pro}$ activity in loss-of-signal assay

To account for mutations that decrease M$^{pro}$ activity and therefore slow down the generation of fluorescent signal, we measured every 12 hours up to 84–96 hours post infection. Over time, the signal increased until a plateau (S9A Fig). To allow continuous measurements, we could not remove the medium covering cells. Instead, we used non-fluorogenic medium (S9B Fig). We then quantified the kinetics of the different mutants with the half maximum time until a signal plateau was reached (TM$_{50}$) (Figs 3C and 3D, and S9E–S9I). We then plotted the resistance and kinetic phenotypes to assess which combinations of mutations are both resistant and could be viable in actual SARS-CoV-2 M$^{pro}$ (Fig 3E).

## VSV-M$^{pro}$ selected mutations are also resistant in a non-infectious SARS-CoV-2 replicon system

To support the resistance phenotypes that have been observed in the VSV-based systems, we used a previously described SARS-CoV-2 replicon assay [46]. By removing essential genes from SARS-CoV-2, the replicon allows testing authentic viral phenotypes such as responses to inhibitors or the effect of mutations, without the risk of genetically altering a dangerous pathogen (Fig 4A).

The mutations selected by VSV-M$^{pro}$ also conferred resistance in the SARS-CoV-2 replicon against nirmatrelvir and ensitrelvir (Fig 4B and 4C). Overall, the resistance phenotypes correlated from moderately to highly, nirmatrelvir showing more correlation than ensitrelvir (Fig 4D). Although the patterns of resistance against nirmatrelvir were amplified in the replicon, they were in general similar with the notable exception of S144A (Fig 4E). Likewise, S144A displayed a stronger phenotype against ensitrelvir in the replicon compared to M$^{pro}$-Off (Fig 4F).

## Purification and resistance characterization of M$^{pro}$ variants through dose response experiments via a fluorescent-based biochemical assay

To further support our findings, recombinant proteases of several selected mutants (WT, L57F, L167F/P168S, L167F/P168S/L57F and Omicron, O/A206T, O/A206T/T198I) were

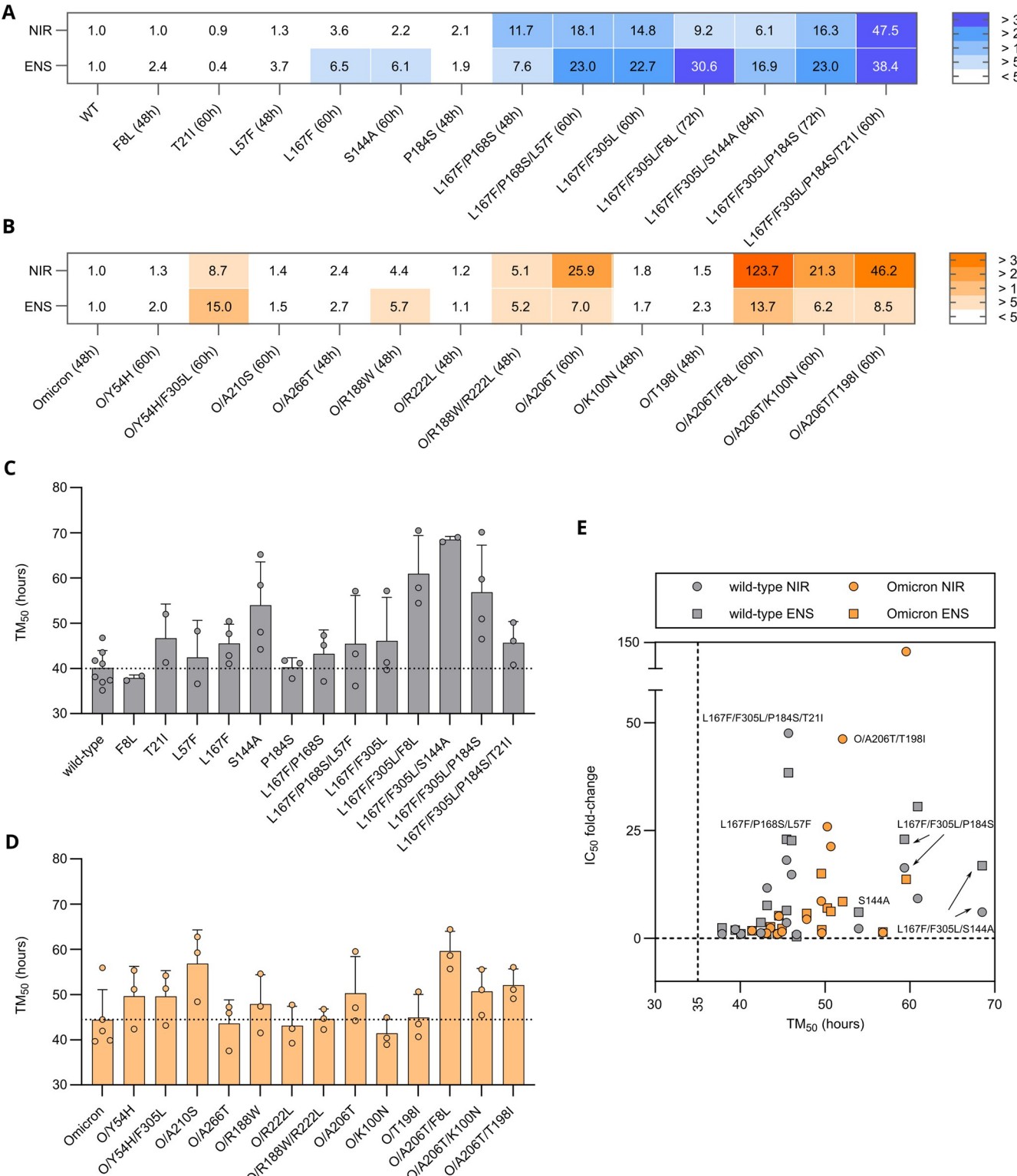

**Fig 3. Resistance data and viral replication kinetics of WT, Omicron and mutant main proteases (A)** $IC_{50}$ fold-changes of nirmatrelvir and ensitrelvir tested against WT-derived mutants. Fold-changes were calculated by dividing the $IC_{50}$ of each mutant by the $IC_{50}$ of the WT protease. Data is displayed as mean of n = 2 / n = 3 / n = 4 independent experiments. **(B)** $IC_{50}$ fold-changes of nirmatrelvir and ensitrelvir tested against Omicron-derived mutants. Fold-changes are calculated by dividing the $IC_{50}$ of each mutant by the $IC_{50}$ of the Omicron protease. Data is displayed as mean of n = 2 / n = 3 independent experiments. **(C)** Bar plot showing the $TM_{50}$ (hours) of WT and derived mutants. Data is displayed as mean ± SD of n = 2 to n = 4 independent experiments

(with the exception for WT with n = 8, used as internal control in each experiment). Each kinetic experiment comprised n = 6 / n = 8 or n = 12 biologically independent replicates. The dotted line represents the average TM$_{50}$ of the WT M$^{pro}$. (**D**) Bar plot showing the TM$_{50}$ (hours) of Omicron and derived mutants. Data is displayed as mean ± SD of n = 2 / n = 3 independent experiments (with the exception for Omicron with n = 5). Each kinetic experiment comprised n = 8 or n = 12 of biologically independent replicates. The dotted line represents the average TM$_{50}$ of the Omicron M$^{pro}$. (**E**) Scatter plot showing the relationship between acquired resistance against nirmatrelvir and ensitrelvir and the change in replication kinetic. Mean TM$_{50}$ values and mean IC$_{50}$ fold-changes are plotted on the x- and the y-axis, respectively (WT, grey; Omicron, orange; circles, nirmatrelvir; squares, ensitrelvir).

produced as shown in **S10A and S10B Fig.** The activity of the purified WT-derived mutant proteases was tested with the recombinant proteases and a reporter bearing the nsp4/5 or N-terminal M$^{pro}$ auto-cleavage sequence of the SARS-CoV-2 polyprotein (**S10C Fig**). The amount of the proteolytically processed fragment of the reporter in a western blot decreased with each added mutation (**S10D Fig**). Additionally, we tested the WT and catalytically inactive M$^{pro}$ C145A with nirmatrelvir and ensitrelvir in a fluorescence cleavage-based assay, where the activity of the purified enzyme leads to an increased release of fluorescent 7-Amino-4-Methylcoumarin (AMC), whereas inhibition blunts it (**S10E Fig**). We also applied an *in vitro* substrate cleavage assay combined with HPLC-detection that facilitated higher substrate doses necessary for kinetic experiments (**S10F Fig**). With the AMC-based assay, we observed increasing resistance phenotypes of WT single to triple mutants against nirmatrelvir and ensitrelvir (**Table 1** and **S11A Fig**). Omicron M$^{pro}$ and derived mutants did not show a clear resistance in this assay (**Table 1** and **S11B Fig**). Kinetic *in vitro* HPLC-based assay data demonstrated that the turnover number k$_{cat}$ of the WT M$^{pro}$ was more affected by the L167F/ P168S mutations than by the L57F mutation (**Table 1**, and **S11C and S11D Fig**). In contrast, the catalytic efficiency (k$_{cat}$/K$_{M}$) decreased significantly for WT as well as for Omicron M$^{pro}$ upon introduction of additional mutations due to the impact of K$_{M}$ (**Table 1**).

### Computational analyses reveal destabilization of protease-inhibitor complex formation and provide insights on protease stability and dimerization affinity upon mutation

To provide potential mechanisms of resistance, we performed molecular modelling with Bioluminate [47–50]. This software uses already existing structural data to calculate the impact of a mutation on the stability of a specific protein conformation. The software returned models of mutant M$^{pro}$ structures and delta stability values (Δ_Stability in kcal/mol) indicating whether a mutation stabilizes (negative values) or destabilizes (positive values) the investigated M$^{pro}$ conformation. Many of the investigated mutations returned positive values, indicating destabilization of both the nirmatrelvir-binding and the nsp5/6 binding conformations in the mutant WT structures (PDB entries 8DZ2 [51] and 7DVW [52]) (**S11E Fig**). The destabilization of the overall structure by mutations correlated with the destabilization of the interaction between M$^{pro}$ and the inhibitor nirmatrelvir as well as M$^{pro}$ and one of its natural substrates (the nsp5/6 junction), suggesting that resistance is often associated with lower proteolytic activity and therefore slower replication.

Nirmatrelvir is a covalent peptidomimetic inhibitor, whereas ensitrelvir a non-covalent, non-peptide inhibitor. Although they both bind to the active site of M$^{pro}$, they locate to different subsites (Nirmatrelvir–S1, S2 and S4; Ensitrelvir–S1, S2 and S1') and these distinct binding modes are affected differently by mutations such as L167F (**Fig 5A**) [51]. Indeed, when modelling the mutant L167F, we obtained a higher Δ_Stability value for ensitrelvir (+235.7 kcal/mol) than for nirmatrelvir (+61.9 kcal/mol).

A structurally related variant is the triple mutant L167F/F305L/F8L, which we investigated in more detail, as both N- and C-terminal regions are mutated (F8L and F305L, respectively).

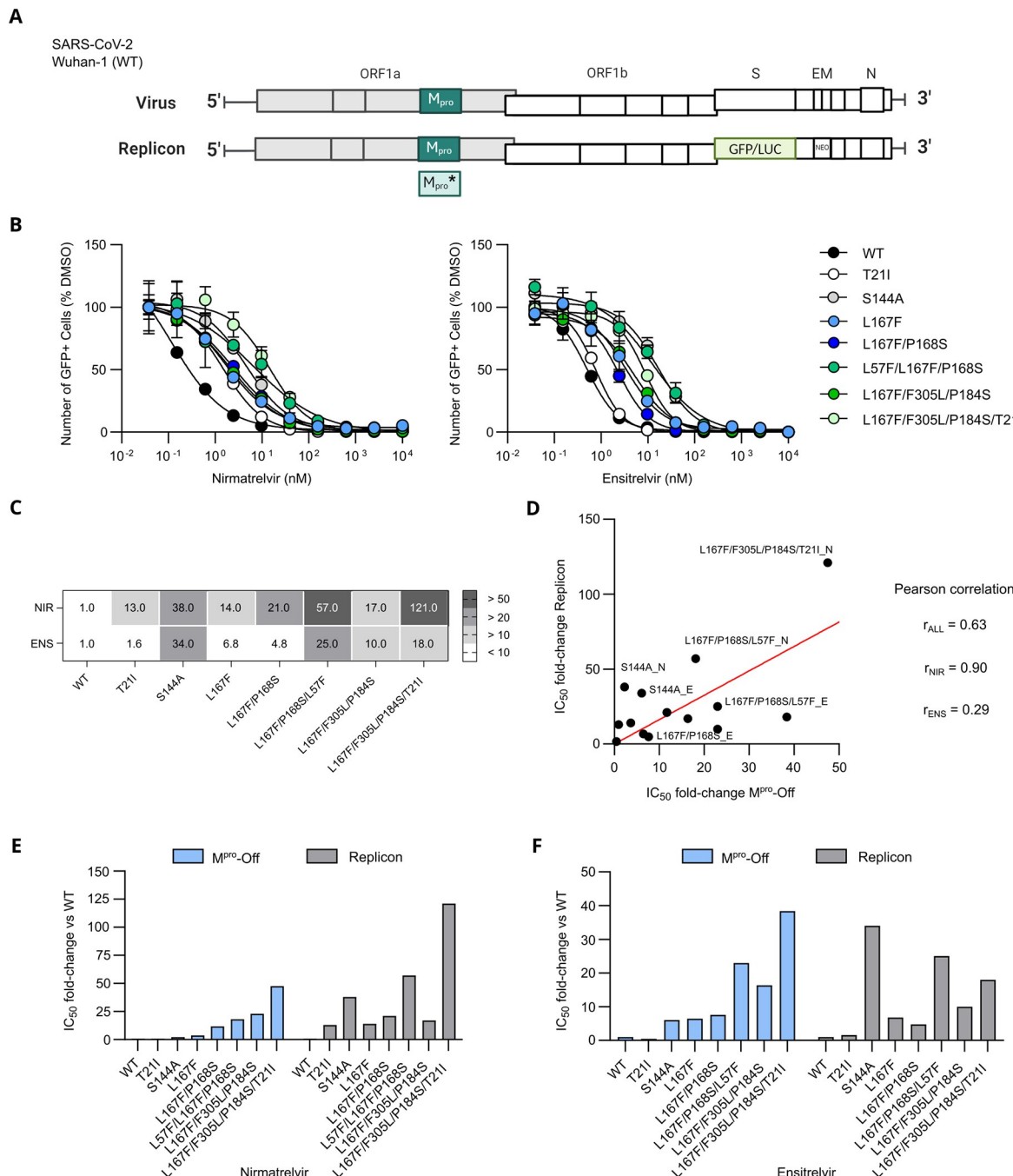

**Fig 4. SARS-CoV-2 replicon dose responses of WT and mutants against nirmatrelvir (A)** Schematic view of the SARS-CoV-2 genome and the replicon system that was generated. Spike, E and M genes were exchanged with GFP/LUC and neomycin-resistance gene (NEO) (**B**) Dose responses experiments of nirmatrelvir (left) and ensitrelvir (right) against M$^{pro}$ mutants tested with the replicon system. Data is presented as ± SEM of n = 3 biologically independent replicates. (**C**) SARS-CoV-2 replicon IC$_{50}$ fold-changes compared to WT. (**D**) Linear regression analysis of resistance data generated by performing M$^{pro}$-Off and SARS-CoV-2 replicon assays. M$^{pro}$-Off IC$_{50}$ fold-changes are plotted on the x-axis and replicon IC$_{50}$ fold-changes are plotted on the y-axis. Dots are labelled with the variant's name, followed by either N or E, which stand for nirmatrelvir and ensitrelvir, respectively. Pearson's correlation coefficients are displayed on the right side of the plot (nirmatrelvir and ensitrelvir grouped: r$_{ALL}$ = 0.63; nirmatrelvir only: r$_{NIR}$ = 0.90, ensitrelvir only: r$_{ENS}$ = 0.29). A strong correlation was found for nirmatrelvir, whereas a low correlation was seen for ensitrelvir. Overall, the two systems have a moderate/high correlation. (**E**) Bar plot of the comparison between the M$^{pro}$-Off assay and replicon resistance phenotypes (IC$_{50}$ fold changes) of nirmatrelvir (**F**). Bar plot of the comparison between the M$^{pro}$-Off assay and replicon resistance phenotypes (IC$_{50}$ fold changes) of ensitrelvir.

**Table 1. Purified M$^{pro}$ dose response and kinetic data.** FC = IC$_{50}$ fold-change vs the parental protease. Nir = nirmatrelvir; Ens = ensitrelvir. Higher V$_{MAX}$ values for O/A206T and O/A206T/T198I are due to a higher enzyme concentration.

| M$^{pro}$ | FC Nir | FC Ens | V$_{MAX}$ [μM*min$^{-1}$] | K$_M$ [μM] | k$_{cat}$ [min$^{-1}$] | k$_{cat}$/K$_M$ [μM$^{-1}$*min$^{-1}$] | M$^{pro}$ conc. [μM] |
|---|---|---|---|---|---|---|---|
| wild type (WT) | 1 | 1 | 74.4 | 329.8 | 37.2 | 0.1128 | 2 |
| L57F | 1.6 | 4.0 | 76.5 | 528.1 | 38.2 | 0.0724 | 2 |
| L167F/P168S | 8.3 | 8.6 | 32.8 | 410.1 | 16.4 | 0.0400 | 2 |
| L167F/P168S/L57F | 20.4 | 57.7 | 24 | 648.7 | 12 | 0.0185 | 2 |
| Omicron | 1 | 1 | 426.6 | 356.8 | 213.3 | 0.5978 | 2 |
| O/A206T | 2 | 1.2 | 2526 | 2467 | 252.6 | 0.1024 | 10 |
| O/A206T/T198I | 2.3 | 1.8 | 1966 | 1885 | 196.6 | 0.1043 | 10 |

The folding of M$^{pro}$ causes the residues F8 and F305 to be close one another, interacting with a π-π T-stack. In our M$^{pro}$ assays, mutations F305L and F8L alone did not confer resistance (**S6 and S7 Figs**), and in our previous work the F305L mutant showed an increase in the replication kinetics [35]. Given the intrinsic structural flexibility of the C-terminal region [53], two orthogonal computational analyses have been carried out to investigate these mutants. Specifically, a first static approach was used to assess the effect of said mutations on the dimerization affinity, while a second dynamic approach was employed to examine the mutant's conformational landscape. First, we performed dimerization affinity prediction calculations using Osprey 3.3 [54] to investigate the effect of the F305L mutation on M$^{pro}$. This program exploits rotamers of residue side chains to provide an estimation of binding affinity between two interaction partners, with higher Log$_{10}$ K* scores indicating stronger and lower Log$_{10}$ K* scores weaker binding. To assess the change in dimerization affinity, we subtracted the Log$_{10}$ K* of the parental protease from the Log$_{10}$ K* of the mutated protease for each residue. A Log$_{10}$ K* score of +9.36 in the F305L mutant was calculated compared to L167F alone, suggesting an increase of M$^{pro}$ dimerization affinity and formation of the mature dimer. A similar value of +9.41 was observed upon further addition of F8L. Upon mutation to L305, the shorter leucine side chain is located further away from the backbone of I152 on protomer A and S123 on protomer B, potentially reducing the electronic repulsion between protomers A and B (**Fig 5C**).

To complement the dimerization affinity analysis based on Osprey, the effect of the simultaneous mutations F8L and F305L mutation was also investigated through molecular dynamics simulations. This mutant exhibited an increase in C-terminus flexibility compared to the wild-type protease, allowing the C-terminus to move more frequently toward the active site (**Fig 5B and S6 Movie**). As can be seen in the video, the C-terminal reaches the S1 subpocket and interacts transiently with the ligand, contributing to the destabilization of its binding mode, eventually leading to its detachment from the binding site. This behavior that we refer to as 'wiper effect', potentially leads to an increased turnover of cleaved peptide substrates and thereby facilitates replication in the presence of an inhibitor [54].

A comparison of recent M$^{pro}$ crystal structures harboring mutations in the domain II-III linker (A193P, E166V, L167F) and the Omicron signature mutation (P132H) revealed high structural variation in this region (**Fig 5D**). These conformational changes seem to allosterically affect the S1', S2 and S4 pockets, in which several of the mutants selected in this study are located. Mutations P168S, P184S from the triple and quadruple variants WT-L167F/P168S/L57F and WT-L167F/F305L/P184S/(T21I) as well as T198I and A206T from O-A206T/T198I are located in or near the domain II-III linker, affecting the conformation of the linker and adjacent binding pockets.

Triple mutant L167F/P168S/L57F was mildly more resistant to ensitrelvir than nirmatrelvir. Structural modelling as well as the proximity of F57 to the S1' pocket suggest that L57F

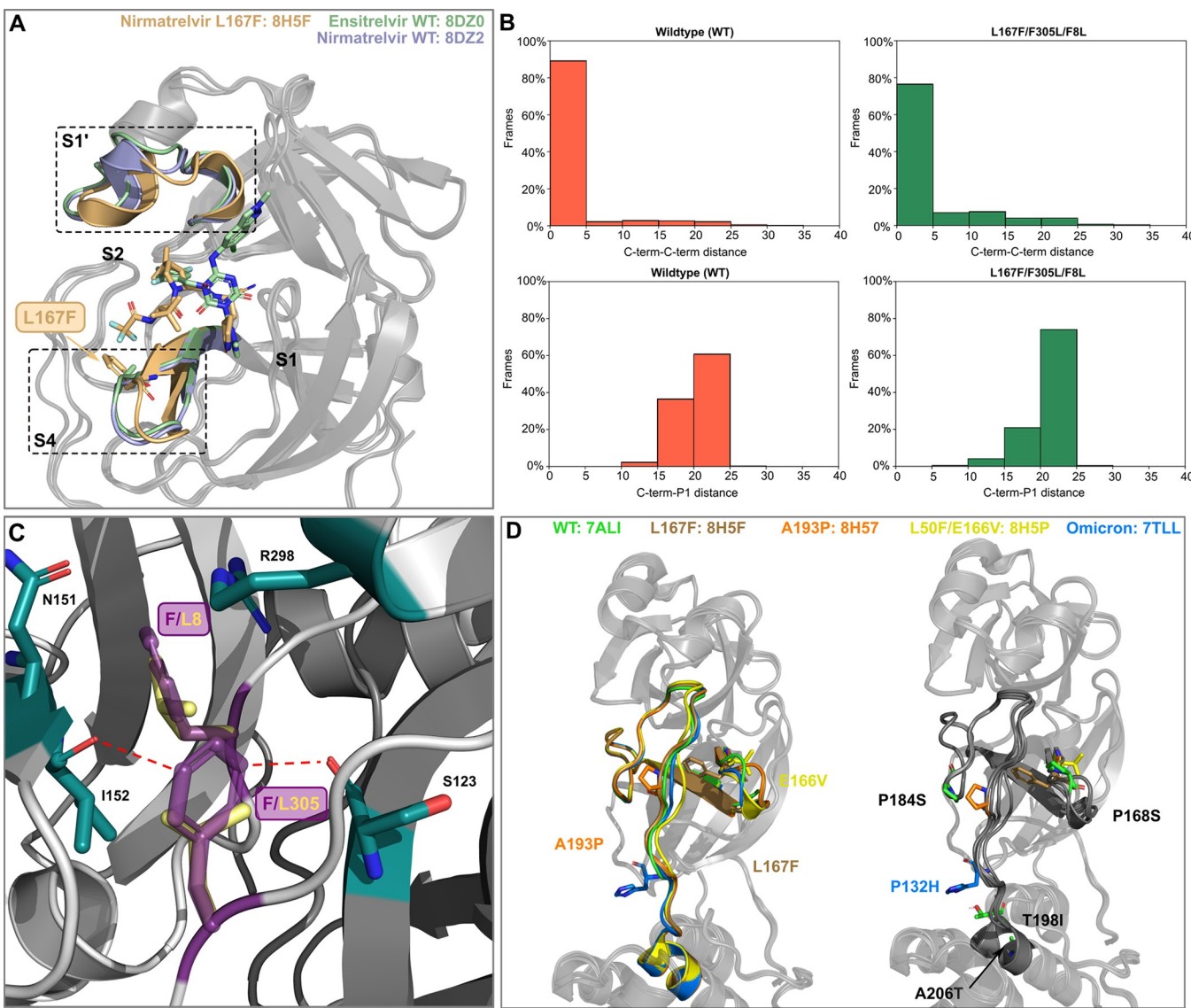

**Fig 5. Structural analysis of mutants.** (**A**) Superposition of nirmatrelvir-WT (8DZ2), ensitrelvir-WT (8DZ0 [51]) and nirmatrelvir-L167F bound (8H5F [55]) crystal structures. L167F is colored in light brown and the two L167F-affected catalytic subpockets are highlighted with dotted-boxes (**B**) Bar plots showing the percentage of frames against C-term/C-term distance or C-term/P1 distance for WT (left) and for the L167F/F305L/F8L mutant (right). (**C**) The F8 and F305 aromatic side chains (dark violet) form a π-π T-stack and F8 in addition interacts with N151, I152, and R298 (green sticks, PDB entry 7ALI [56]). F8 is located within the homodimer interface. The conjugated p-orbitals of the F305 side chain are located nearby of the I152 and S123 backbone oxygen atoms, potentially leading to electronic repulsion. In the F305L mutant, the distance of the leucine side chain (yellow sticks) to these oxygen atoms is considerably increased. (**D**) Superposition of recent Mᵖʳᵒ and Omicron crystal structures harboring different mutations (A193P (PDB entry: 8H57 [55]), or near (E166V (PDB entry: 8H5P [55]), L167F (PDB entry: 8H5F [55]) in the domain II-III linker loop, and the Omicron signature mutation (P132H (PDB entry: 7TLL [57]) (left panel). All residues that were found to alter the inhibitor binding and are also located on the domain II-III linker (T198I) or nearby (P168S, P184S and A206T) are mapped onto the structure in the right panel.

destabilizes ensitrelvir binding preferentially. Stability calculations on L167F/P168S/L57F resulted in a large destabilization for nirmatrelvir (235.7 kcal/mol) and even stronger destabilization of ensitrelvir (417.4 kcal/mol).

O/A206T was frequently observed in our selection experiments, prompting us to investigate it in detail. The larger, polar side chain of the threonine mutant is not compatible with the hydrophobic local environment, sterically clashing with the residues nearby and requiring

**Table 2. MS coefficients of WT and mutants.** MS coefficients were determined for each TTMD replicate. The rank is based on inhibitor-protein complex stability. The lower the rank, the higher the stability of the complex. The average MS value was calculated from the replicates as reported in the original work [58] and it is further described in the methods section. Stabilities were ranked increasingly from the highest (1), which was the Wuhan-1 WT.

| | Rank | MS score | RUN 1 | RUN 2 | RUN 3 | RUN 4 | RUN 5 |
|---|---|---|---|---|---|---|---|
| **Nirmatrelvir** | | | | | | | |
| wild type (WT) | 1 | 0.0037 | 0.0029 | 0.0028 | 0.0048 | 0.0037 | 0.0047 |
| L167F/F305L/F8L | 2 | 0.0042 | 0.0046 | 0.0041 | 0.0038 | 0.0039 | 0.0083 |
| L167F/F305L/S144A | 3 | 0.0044 | 0.0029 | 0.0044 | 0.0052 | 0.0042 | 0.0047 |
| L167F/F305L/P184S/T21I | 4 | 0.0045 | 0.0040 | 0.0044 | 0.0052 | 0.0049 | 0.0038 |
| L167F/F305L/P184S | 5 | 0.0051 | 0.0049 | 0.0071 | 0.0050 | 0.0054 | 0.0026 |
| L167F/P168S/L57F | 6 | 0.0056 | 0.0055 | 0.0069 | 0.0111 | 0.0038 | 0.0045 |
| **Ensitrelvir** | | | | | | | |
| wild type (WT) | 1 | 0.0026 | 0.0024 | 0.0019 | 0.0029 | 0.0026 | 0.0043 |
| L167F/P168S/L57F | 2 | 0.00315 | 0.0033 | 0.0028 | 0.0046 | 0.0023 | 0.0033 |
| L167F/F305L/P184S | 3 | 0.00388 | 0.0037 | 0.0041 | 0.0040 | 0.004 | 0.0032 |
| L167F/F305L/S144A | 4 | 0.00389 | 0.0061 | 0.0033 | 0.0047 | 0.0037 | 0.0027 |

conformational changes to accommodate the side chain of threonine, affecting both domain III and the domain II-III linker (residues 185–200) (**Fig 5D**) [52].

To investigate the protease-ligand complex stability while considering protein dynamic effects, we performed Thermal Titration Molecular Dynamics (TTMD) [58] simulations. These simulations return the so-called MS coefficient, which is a measure for how long the protein-ligand complex is stably formed. This measure facilitates a qualitative comparison of the different protein-ligand complex stabilities, for example, nirmatrelvir bound to different mutant M^pro variants.

Five catalytic site mutants (**Figs 2A and 3A**) that arose from the WT-L167F-M^pro were selected for this modelling and compared to WT M^pro-nirmatrelvir. Three M^pro-ensitrelvir complexes were modelled (**Table 2**). Each TTMD simulation was repeated five times and an average calculated from these *in silico* replicates. To illustrate the stabilities of the different complexes, we ranked the mutants according to MS coefficients, showing that the nirmatrelvir-WT complex is the most stable, followed by L167F/F305L/F8L > L167F/F305L/S144A > L167F/F305L/P184S/T21I > L167F/F305L/P184S and > L167F/P168S/L57F. TTMD simulations were also performed on ensitrelvir-M^pro complexes, were again we observed that the ensitrelvir-WT complex is the most stable, followed by L167F/P168S/L57F > L167F/F305L/P184S > L167F/F305L/S144A.

TTMD simulations and relative trajectory analysis generate further useful metrics that can be visualized, most importantly titration timeline plots, root mean square deviation (RMSD) plots, titration profiles, interaction energy plots and inhibitor-protease structures. Interaction energy plots and inhibitor-protease structures of Wuhan-1 WT M^pro show that the interaction pattern (per-residue interaction energy) changes and the active site constantly reshapes throughout the simulation (**Fig 6A** and **S1 Movie–right panel**). The active site reshaping is particularly strong in the most resistant mutant structures L167F/F305L/P184S/T21I and L167F/P168S/L57F in the area affected by the mutations (**Figs 6A–6C, and S12A–S12D,** and **S1** and **S5 Movies**). As displayed in the titration timeline plot and the titration profile of the WT M^pro, the interaction with nirmatrelvir became unstable at 80 ns, 380 K (**S12A Fig**). These metrics help to assess the dynamic instability of mutated M^pro variants relative to the WT. In mutant L167F/F305L/P184S/T21I the TTMD indicated a stabilizing effect of the T21I mutation (**Fig 6B** and **S5 Movie**). L167F/P168S/L57F led to a loss of interaction with the backbone

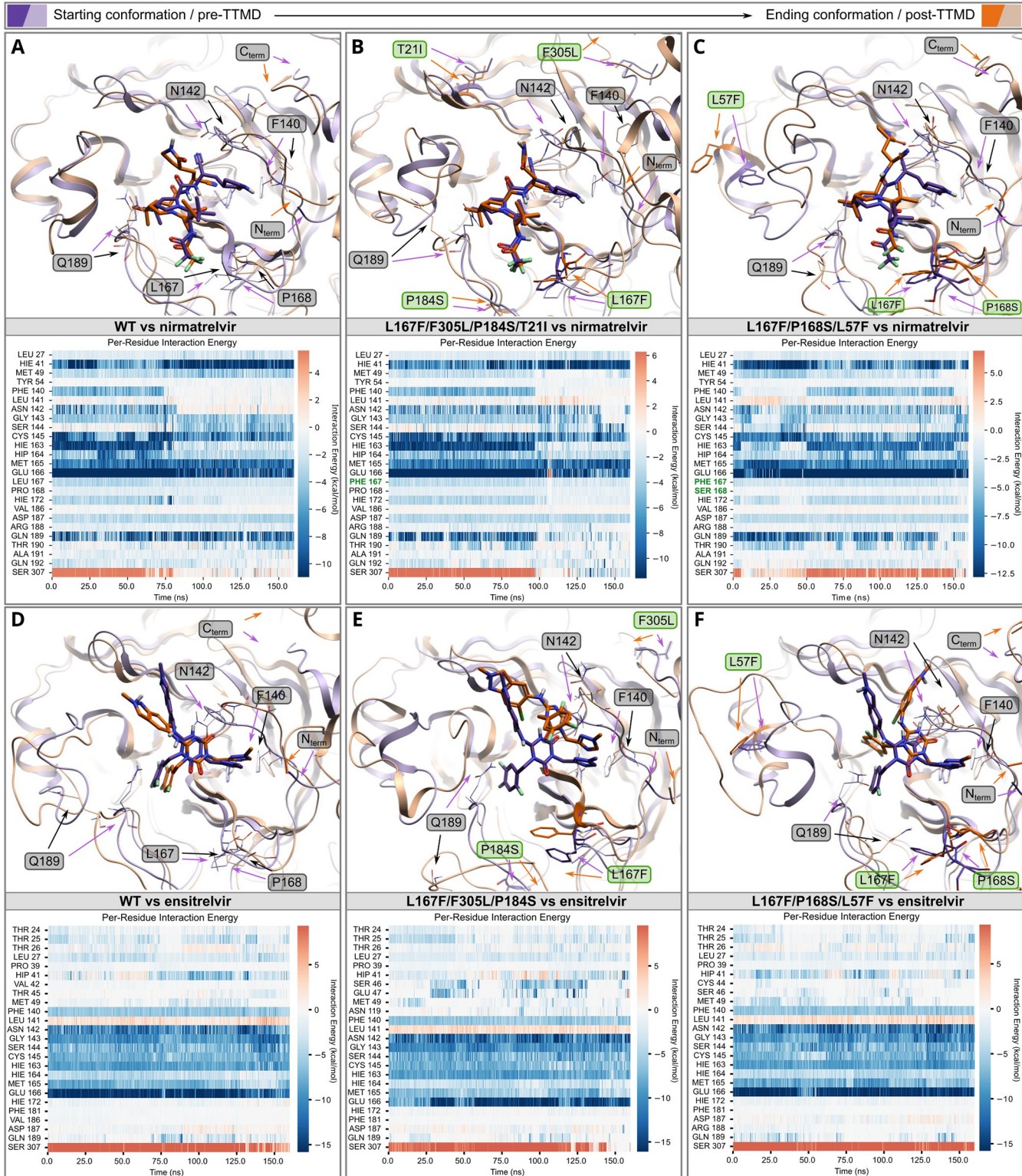

**Fig 6. Thermal Titration Molecular Dynamics (TTMD) experiments of nirmatrelvir and ensitrelvir M^pro complexes.** TTMD simulation data of the nirmatrelvir-WT-M^pro (**A**), nirmatrelvir- L167F/F305L/P184S/T21I-M^pro (**B**) and nirmatrelvir- L167F/P168S/L57F-M^pro (**C**) complexes. Top: overlay of M^pro structure at the beginning (violet/purple) and at the end of the simulation (light brown/orange); bottom: heat map of interaction energies between ligand and surrounding residues. TTMD simulation data of the ensitrelvir-WT-M^pro (**D**), ensitrelvir- L167F/F305L/P184S-M^pro (**E**) and ensitrelvir- L167F/P168S/ L57F-M^pro (**F**) complexes. Top: overlay of M^pro structure at the beginning (violet/purple) and at the end of the simulation (light brown/orange); bottom: heat

map of interaction energies between ligand and surrounding residues. Mutated residues and residues that are mentioned in the results/discussion are labelled in green and grey, respectively.

of crucial catalytic site residues (164–166) by altering the floor of the binding site, causing a drag effect on nirmatrelvir (**Fig 6C and S1 Movie**). The resistance mechanism of the L167F/F305L/S144A mutant combines an increased instability of the oxyanion loop (residues 138–145) caused by S144A. This potentially increases plasticity of the floor of the binding site due to the L167F mutation, which finally leads to the loss of the hydrogen bond between nirmatrelvir and H163, subsequently resulting in the loss of interaction with residue Q189 [53] (**S12B Fig and S2 Movie**). L167F/F305L/F8L has only one mutation within the catalytic site (L167F) and two other ones located at the dimeric interface, F8L and F305L (**S12D Fig and S3 Movie**). As mentioned above, the C-terminus is freer to move in the available space, potentially competing with the ligand in a 'wiper'-like motion (**S6 Movie**). Ensitrelvir binds to a different subsite of the catalytic pocket of M^pro, which is reflected in its interaction energy plot (**Fig 6D–6F**). In L167F/F305L/P184S, the P184S mutation enhances the plasticity of the loop spanning from residue S184 to Q192, thereby reducing the ligand-pocket shape complementarity, which is essential for both nirmatrelvir and ensitrelvir positioning. The higher plasticity of the loop affects the stability and the number of water molecules hosted between domains I-II, including the stable water commonly interacting and connecting H164, H41, and D187 residues, a water molecule that holds not only a structural role but has also been described as a third member of the catalytic dyad (**Fig 6E**). The mutations L167F/P168S/L57F and L167F/F305L/S144A reshaped the catalytic pocket and strongly destabilized ensitrelvir interactions in the S2 pocket, leading to a loss of the hydrogen bond with E166 and the π-π stacking with H41 (**Figs 6F and S13D**).

## Discussion

In this study, we used a previously developed, safe, VSV-based tool to select for protease inhibitor resistance mutations. We demonstrated the capability of our selection tool to achieve SARS-CoV-2 M^pro variants with multiple substitutions and potential Omicron-related mutants relevant to the current pandemic situation. We confirmed the resistance of several of our mutations in biochemical and SARS-CoV-2 replicon assays. Lastly, we provide comprehensive static and dynamic modelling to suggest resistance mechanisms.

### Focus on specific mutants

First, we generated a large number of mutants, which we subsequently filtered on a three-parameter basis for further characterization: proximity to the inhibitor, prevalence in the GISAID database, and frequency of a specific mutation in our selection experiments. Substitutions that appeared close to the inhibitor are likely to yield meaningful, structural explanations that can then inform structural derivatization of inhibitors. Highly frequent substitutions in GISAID are relevant because they occur in viruses that can efficiently propagate in humans and therefore should not be detrimental to the enzymatic activity. Lastly, mutations that occur frequently in selection experiments could be favoured routes of evolution for the M^pro enzyme [59] and thus the mechanism might be of interest.

### Wuhan-1 (WT) M^pro resistance mutations

Based on the VSV-L167F-M^pro variant, additional substitutions close to the inhibitor were selected, namely S144A, P184S and L57F. In some of those mutants, we observed differences in the susceptibility to the two protease inhibitors nirmatrelvir and ensitrelvir. Interestingly,

the mutant L167F/F305L/F8L demonstrated a considerably greater degree of resistance to ensitrelvir than to nirmatrelvir. Both L167F/F305L/P184S and L167F/F305L/L57F showed a similar trend. L167F/P168S/S144A had a comparable resistance profile against both protease inhibitors. Mutations of S144 can simultaneously cause an increase in drug resistance and a decrease in the catalytic activity of the protease. Consistently, in our kinetic experiments and in the literature [18] this mutation impaired replication, especially in the triple-mutant.

As mentioned above, the second filtering parameter we applied was the absolute frequency of a specific nsp5/M^pro substitution in the GISAID database. Residues in catalytic sites tend to be more conserved [41,42], and we expected them to not be frequently represented. Indeed, most highly-frequent mutants were observed in positions further away from the active pocket. P184S is an exception as it is frequently represented (4713 entries, as of 01/06/2023) and is near the catalytic pocket. During the rescue (generation of a virus from a plasmid) of L167F/F305L/P184S, one sample had acquired an additional mutation, namely T21I. The resistance profile of L167F/F305L/P184S changed from more resistant against ensitrelvir to stronger resistance against nirmatrelvir upon introduction of the T21I mutation. However, T21I alone neither conferred resistance against nirmatrelvir nor ensitrelvir in both our gain- and loss-of-signal assays. In addition to a change in the resistance profile, the T21I mutation in the quadruple mutant caused a considerable decrease in the $TM_{50}$. Consistently, it has been reported in the literature as a compensatory mutation for the fitness loss caused by other resistance mutations, such as E166V [18]. T21I could support or compensate the increased plasticity of L167F/F305L/P184S we found in TTMD simulations by an allosteric, re-stabilizing effect [25,60].

Taken together, it seems that L167F, P168S, S144A and P184S mutations deform the binding pocket and loosen the electrostatic interactions between nirmatrelvir and the protease. This observation indicates that not only mutations of residues directly interacting with the inhibitor can decrease susceptibility, but mutations of residues within the catalytic pocket can trigger conformational changes in the enzymatic cleft that impact in the inhibitor-protease complex formation as well, a finding that is also supported by recent structural data [55].

The triple mutants L167F/F305L/F8L and L167F/P168S/L57F are among our most resistant M^pro variants and showed the most marked difference in susceptibility between the two inhibitors nirmatrelvir and ensitrelvir. The residues F305 and F8 are in close contact with each other in the folded protein and this is expected to be a stabilizing interaction for the dimer interface. However, other residues are viable at this position, for example leucine (L) has been shown to be the preferred residue instead of phenylalanine (F) [42]. In fact, we calculated an increase in the dimerization affinity from F305 to L305, suggesting an improved formation of the fully functional dimeric protein upon mutation. We hypothesize that the simultaneous mutation of both F305 and F8 residues into leucines results in a modified interaction between the C-term of the first protomer and the N-term of the second, increasing the conformational degree of freedom of the C-term, thereby leading to a "wiper" mechanism that could compete with the inhibitor for the catalytic site.

## Omicron-M^pro resistance mutations

The Omicron-M^pro sequence has a signature substitution at amino acid position 132, exchanging proline for histidine (P132H). This mutation has been shown to not confer resistance to approved protease inhibitors, such as nirmatrelvir and ensitrelvir [30–33], in agreement with our findings, but altered thermal stability [34]. To reduce biosafety risks, previous studies based on authentic SARS-CoV-2 to select resistance mutations have used early variants of SARS-CoV-2 [18,19,21]. Protease inhibitor resistant mutants derived from such early variants would unlikely be able to compete with Omicron variants if released accidentally. This caveat

does not apply to mutation selection performed with our VSV-M$^{pro}$ system. Consequently, we selected protease inhibitor resistant Omicron M$^{pro}$ mutations within and outside the catalytic site. The catalytic site mutant O/Y54H, which occurred 10 times independently, and O/R188W showed only mild resistance alone as well as in combination with R222L. O/Y54H/F305L showed an increased preferential resistance against ensitrelvir, with a 15-fold-change of the IC$_{50}$ against ensitrelvir in the loss-of-signal assay (M$^{pro}$-Off). The mutants K100N, T198I, A210S, and A266T were frequent in GISAID, therefore we picked them for resistance characterization. Double mutants O/A206T/F8L and O/A206T/T198I showed strong selective resistance against nirmatrelvir (up to ~125-fold-change) in the M$^{pro}$-Off system, while being less effective for ensitrelvir.

Most Omicron and WT substitutions, however, occurred in the amino acid sequence that constitutes domain III (residues 200–306) of M$^{pro}$. We chose to characterize the A206T mutation due to the unusually high frequency in the selected VSV-O-M$^{pro}$ pool of mutants. A206T only fits our third filtering parameter: frequent selection in our experiments, as it is far from the catalytic site and scarcely represented in the GISAID database. Likewise, some mutations may confer an advantage for viral replication only in the presence of a protease inhibitor but could be unfavourable in an untreated individual. In a study on SARS-CoV M$^{pro}$ polyprotein maturation [61], the authors proposed that despite the presence of deleterious mutations that hinder mature dimerization, M$^{pro}$ retains its ability to cleave its N-terminal cleavage site owing to a weak, immature dimerization between two polyprotein monomers (1 and 2) catalysed by domain III (1)–domain III (2) interaction. We speculate that mutations of residues between 200 and 306, most frequently found in VSV-O-M$^{pro}$, positively contribute to this interaction.

## Viral replication kinetics

In kinetic studies, we observed that the gain in signal was slower for some mutants, which might be an indirect read-out of M$^{pro}$ activity. The L167F/F305L/S144A mutant was among the slowest variants and the time it took to reach the plateau was 28.5 hours higher longer than for the WT M$^{pro}$. However, mutations that were described as compensatory in the literature, such as T21I [18], could partially reverse the kinetic attenuation. T21I appeared after generating VSV-L167F-F305L-P184S-M$^{pro}$ from its plasmid (also called "rescue"), and R222L after plaque purification of VSV-O-R188W-M$^{pro}$. T21I and R222L are likely compensating for substitutions that occur in the catalytic site. For example, T21I seems to compensate for the loss of replicative fitness of the previously described, highly resistant M$^{pro}$-E166V variant [18]. Furthermore, T21I is a very frequently represented substitution in the GISAID database (21248 entries, 1$^{st}$ of June 2023). From mutations generated in the VSV-O-M$^{pro}$ variant, viral replication kinetics exhibited a different behaviour in comparison to VSV-L167F-M$^{pro}$ variants. Overall, mutations related to the Omicron-M$^{pro}$ did not decrease the viral replication rate to the same extent as in the L167F-M$^{pro}$. The slowest variant was O/A206T/F8L, with an increase of ~15 hours to reach the plateau. As expected, most mutations that have been computationally predicted to decrease protease stability indeed had a negative impact on viral replication, with higher TM$_{50}$ values related to M$^{pro}$ variants bearing such mutations. Furthermore, they were less represented in isolates in the GISAID repository (1 or 2 digit entries), whereas those that had similar TM$_{50}$ values compared to the WT such i.e. F8L, P184S or K100N and T198I alone, were more frequent (3 or 4 digit entries).

## Phenotype validation with SARS-CoV-2 replicons and enzymatic assays

To validate resistance phenotypes in independent assay systems, we used a previously described SARS-CoV-2 replicon [46] and found that the resistances between the two assay

systems correlated. A notable exception was the mutation S144A, of which the resistance was underestimated by the VSV-based assays. Introducing selected WT-derived mutations into purified M$^{pro}$ enzymes showed that $k_{cat}$ of the WT M$^{pro}$ was more affected by the L167F/P168S mutations than by the L57F mutation. Both L167 and P168 locate at the inhibitor binding site, which also recognizes the C-terminal part of the natural substrate to be cleaved. Thus, mutations may weaken substrate recognition, but once bound, the turnover is fast. Omicron's turnover numbers were comparable in the absence and presence of a T198I and/or A206T mutation. These mutations are located in domain III and likely affect M$^{pro}$ dimer formation.

## Caveats of the study

Chimeric VSV-Spike, where the VSV glycoprotein G was replaced by the SARS-CoV-2 spike, has been used previously to predict antibody escape mutations [62–64]. Before that, VSV was shown many times to be promiscuous in the context of its glycoprotein, which could easily be replaced for example with those of Ebola and Marburg viruses (EBOV, MARV) [65] or lymphocytic choriomeningitis virus (LCMV) [66]. To replace one of the VSV intergenic regions with a protease and thereby use VSV as a protease mutation tool, however, was a previously untapped area of research. Therefore, our resulting mutations need careful evaluation. We observed many mutations in chimeric VSV-M$^{pro}$, some of which aligned exactly with those identified in SARS-CoV-2 gain-of-function experiments reported in the literature [18,19], whereas others were completely different. One explanation may be the artificial system used in our experiments, which uses cis-cleavage, the only requirement for M$^{pro}$ processing in VSV-M$^{pro}$ replication. In contrast, in experiments with authentic SARS-CoV-2, both cis- and trans-cleavage must be preserved in the development of resistance mutations. As described above, during auto-cleavage there is an intermediate dimerization state, different from the mature protease. This intermediate state may require specific interactions that differ from those involved in the mature dimer. Another reason is the difference between the natural polyprotein of coronaviruses and the artificial G-M$^{pro}$-L polyprotein expressed by chimeric VSV-M$^{pro}$. The observed divergence between SARS-CoV-2 and VSV may also be attributed to the differences between two viral polymerases: SARS-CoV-2 RNA-dependent RNA polymerase (RdRp) and VSV polymerase L. It has been reported in the literature that the SARS-CoV-2 polymerase can proofread [67], whereas VSV lacks such mechanisms [37], leading to a much higher error rate of 1/10,000 nucleotides, as described previously [36]. However, this could also be seen as an advantage as resistance mutations may develop faster in the error prone VSV-M$^{pro}$ replication system. Lastly, VSVs replication cycle is much faster than that of SARS-CoV-2, in part owed to the simpler structure of the VSV genome and replication strategy. Therefore, VSV proteins are expressed faster and in greater number in a shorter time, which accounts for the higher amounts of M$^{pro}$ inhibitor needed when treating VSV-M$^{pro}$ vs. SARS-CoV-2.

## Conclusion

Overall, we conclude that most protease inhibitor resistance mutations destabilize the mature conformation of M$^{pro}$ and consequently the inhibitor-M$^{pro}$ complex formation, which finally impairs inhibitor binding. These resistance mutations can be selective either for nirmatrelvir or for ensitrelvir, or impact both. Moreover, the extended use of protease inhibitors will increase the risk of selecting SARS-CoV-2 protease-inhibitor resistant variants. Therefore, to combat such variants, there is still a need for new protease inhibitors that ideally target alternative conformations or regions of the protease [53,68], and for rationale design of new inhibitors less affected by mutations [69].

## Materials and Methods

### Study design

The overall rationale of the study was to use a previously developed mutation selection tool based on VSV to select and characterize a comprehensive collection of SARS-CoV-2 main protease inhibitor escape mutants. The study was performed on cell lines, in biochemical settings and *in silico*, and no animal husbandry or human participants were involved. Human cell lines with replicating BSL-1 and -2 viruses were treated with protease inhibitors to observe resistance phenotypes in appropriate facilities. Viral titres were determined using TCID$_{50}$. Measurement readouts were fluorescence-based, detected by ELISpot/FluoroSpot and multi-well readers. Autofluorescent fibers were excluded automatically from spot counting in the ELISpot readout. Experiments were neither blinded nor randomly distributed to experimenters. We chose sample sizes empirically based on experience from former studies. At least two and up to four biologically independent replicates were performed per condition. In replication kinetic experiments, at least 6 and up to 12 biologically independent replicates were performed. Biologically independent meant distinct wells with the same condition, not multiple measurements of the same wells (technical replicates). M$^{pro}$-Off resistance phenotypes were reproduced at least twice, usually more often and in different combinations.

### Cloning strategies

The chimeric VSV variant with M$^{pro}$ instead of the intergenic region between G and L was cloned as previously described [35]. VSV-G-M$^{pro}$-L carrying the Omicron signature mutation, P132H (VSV-O-M$^{pro}$) was cloned as follows: N-terminal fragment comprising part of G and M$^{pro}$ was amplified by PCR using primers 33n-before-KpnI-for and Omicron rev with VSV-G-M$^{pro}$-L as a template. The C-terminal fragment comprising the remaining part of M$^{pro}$ and part of L was amplified by PCR using primers Omicron for and 33n-after-HpaI-rev. N- and C-terminal fragments were joined by Gibson assembly in a VSV-backbone vector digested with KpnI and HpaI. VSV-G-M$^{pro}$-L L167F (VSV-L167F-M$^{pro}$) was plaque purified as it was generated in a previous study.

M$^{pro}$-Off and -On point mutants were generated by point directed mutagenesis on parental plasmids (GenBank accession codes: M$^{pro}$-Off: ON262565; M$^{pro\ pro}$-On: ON262564) with mutation primers and the Herculase II Fusion DNA Polymerase. Herculase is a polymerase that can overcome difficult (GC-rich) sequences and amplify large plasmids. However, primers could not be chosen for optimal design because the mutation site was fixed. For this reason, this simple point directed mutagenesis did not work for each construct. For M$^{pro}$-On and M$^{pro}$-Off plasmids, where point-directed mutagenesis did not work, we used mutagenic Gibson assembly as previously described [35]. Cloning primers used in this study are shown in **S3 Table**.

### Cell lines

BHK-21 cells (American Type Culture Collection, ATCC) were cultured in Glasgow Minimum Essential Medium (GMEM) (Lonza) supplemented with 10% fetal calf serum (FCS), 5% tryptose phosphate broth, and 100 units/ml penicillin and 0.1 mg/ml streptomycin (P/S) (Gibco). 293T cells (293tsA1609neo, ATCC), and 293-VSV (293 expressing N, P-GFP and L of VSV) [70] were cultured in Dulbecco's Modified Eagle Medium (DMEM) supplemented with 10% FCS, P/S, 2% glutamine, 1x sodium pyruvate and 1x non-essential amino acids (Gibco). Vero E6 (ATCC CRL-1586) were cultured in DMEM supplemented with 5% FCS (VWR) and 1% penicillin–streptomycin–glutamine (PSG) solution (Corning). Huh-7 cells (ATCC) were cultured in DMEM supplemented with 10% FCS.

## Virus recovery ('rescue')

VSV-G-M$^{pro}$-L WT and a few mutants were rescued in 293T cells by CaPO$_4$ transfection of whole-genome VSV plasmids together with T7-polymerase, N-, P-, M-, G- and L expression plasmids as helper plasmids [43]. Briefly, genome and helper plasmids were transfected into 293T in the presence of 10 µM chloroquine to avoid lysosomal DNA degradation. After 6 to 16 hours, chloroquine was removed, and cells were cultured until cytopathic effects occurred. M and G proteins were used as helper plasmids; although these proteins are optional in the recovery of VSV, they were chosen here as a precaution to support the rescue of a potentially attenuated virus variant. After the rescue, viruses were passaged on 293-VSV cells and plaque purified twice on BHK-21 cells. ΔP and ΔL VSV variants expressing dsRed were produced on replication supporting 293-VSV cells. VSV-G-M$^{pro}$-L was fully replication competent and produced on BHK-21 cells.

## Mutation selection assay

10$^4$ BHK-21 cells/well were seeded in a 96-well plate one day before infection with the chimeric VSV-M$^{pro}$ viruses. VSV-L167F-M$^{pro}$ and VSV-O-M$^{pro}$ infected cells (MOI = 0.01) were treated with nirmatrelvir with concentration ranging from 10 to 100 µM. The resistant variants VSV-L167F/P168S-M$^{pro}$ and VSV-L167F/F305L-M$^{pro}$ at MOI = 0.01 were passaged 5 times, and with increasing concentrations of inhibitor (from 50 to 100 µM). VSV-O-M$^{pro}$ and VSV-O/A206T-M$^{pro}$ at MOI ranging from 0.001 and 0.01 were passaged only once with increasing concentrations of inhibitor (from 6 to 100 µM). Each virus variant occupied from 24 wells to 96 of the 96-well plate. Supernatants from wells that displayed cytopathic effect after 48–72 hours were collected for downstream viral RNA isolation, cDNA synthesis, PCR amplification and Sanger or Nanopore sequencing as described in this section.

## Viral RNA isolation, cDNA synthesis and M$^{pro}$ sequencing

VSV-G-M$^{pro}$-L RNA was isolated either by using the E.Z.N.A. Viral RNA Kit (Omega Bio-Tek Inc.), the NucleoSpin RNA Virus (Macherey-Nagel GmbH) or by semiautomated RNA extraction using an Easymag (BioMerieux). BHK-21 cells were infected with the respective VSV-G-M$^{pro}$-L variant in 96-well plates as described above. Virus-containing supernatants were collected from individual 96-wells and the RNA was purified from the supernatants according to manufacturers' instructions (Easymag). Then, cDNA was synthesized from isolated viral RNA by RevertAid RT Reverse Transcription Kit (Thermo Fisher Scientific). G$_{Cterm}$-M$^{pro}$-L$_{Nterm}$ fragment sequence was amplified by PCR with primers (primer_for: CTCAGGTGTTCGAACATCCTCAC and primer_rev: GATGTTGGGATGGGATTGGC) and either sent for Sanger sequencing (MicroSynth AG) or sequenced using Nanopore (as described below). For Nanopore sequenced samples, the fraction of mutated virus of the full population is shown in **S4 Table**. Obtained sequences were mapped to the L167F-M$^{pro}$ or Omicron-M$^{pro}$ reference sequences in Geneious Prime 2023.0.1 and examined for mutations.

## Nanopore sequencing workflow

After the first two selection experiments where we sequenced our samples with Sanger sequencing (MicroSynth AG), we used Nanopore sequencing. First, PCR amplification products of the G$_{Cterm}$-M$^{pro}$-L$_{Nterm}$ fragments, as described above, were used. The sequencing libraries were prepared using the Rapid Barcoding Kit SQK-RBK110.96 (Oxford Nanopore Technologies, ONT), and up to 96 samples were multiplexed and sequenced on a MinION Mk1B sequencer with R9.4.1 flowcells (ONT). Raw data in the form of electrical signals are

translated into nucleotide sequences (base-calling) and saved in fastq files. Basecalling, using the super high accuracy model, demultiplexing, as well as barcode and adapter sequence trimming, was performed in Guppy (version 6.1.5, ONT).

Raw reads were filtered to a PHRED quality score of $\geq$ Q15 and length between 200 bp and 1800 bp using SeqKit (version 2.4.0) [71], aligned to the reference sequence using minimap2 (version 2.22) [72], followed by sorting and indexing using SAMtools (version 1.13) [73]. SAMtools depth was used to check sufficient depth. To find SNVs, we used LoFreq for variant calling with—min-cov option set to 300 (v2.1.3.1) [74].

The resulting VCF files were imported to Geneious Prime 2023.0.1 and called variants manually checked for plausibility.

### Unrooted phylogenetic tree generation and visualization

Mutations had to be manually inserted into the main protease sequence for each variant. The phylogenetic tree was generated using the online tool http://www.phylogeny.fr/ with standard parameters. The output as.newick file was imported to Geneious Prime 2023.0.1, visualized and exported as PDF. Graphic rearrangements and overhauls were made using Inkscape version 1.1.

### Collection of GISAID frequencies of generated mutations

To obtain updated nsp5 frequencies, metadata from the EpiCoV database (as of 1^st June 2023) was used; specifically, the 'all_mutations'field from a JSON dump obtained with GISAID credentials. Using Python 3.9 and the pyarrow and pandas' libraries, sequences were classified into WT (not assigned Omicron lineage, and not carrying the P132H mutation) and Omicron (either assigned Omicron lineage or carrying the P132H mutation). Then, for both these groups of sequences, the counts of 95 nsp5 mutations of interest were aggregated.

### SARS-CoV-2 variant phylogenetic analyses using Ultrafast Sample placement on Existing tRees (UShER)

Phylogenetic trees were generated for the resistance variants of interest using patient derived sequences deposited in the GISAID database. For each variant of interest, viral genomes harboring the corresponding mutation in Nsp5 were retrieved using the GISAID EpiCoV web server after filtering to consider only viruses descending from the Omicron lineage and removing genomes with low sequence coverage. These sequences were subsequently uploaded to the UShER [75] web server to generate phylogenetic trees using all the sequences available in the GISAID database, visualized using the Auspice.us web application and edited to highlight the variants of interest using Adobe Illustrator. The frequency of each resistance mutation within Omicron and Pre-Omicron lineages was similarly determined using the GISAID EpiCoV web server by dividing the number of occurrences of the corresponding Nsp5 mutation within Omicron or all other SARS-CoV-2 lineages by the total number of viral genomes deposited belonging to each respective lineage. Histograms comparing the frequency of the variants within the Omicron and Pre-Omicron lineages were generated using GraphPad Prism 9.

### Screening assay with ELISpot/FluoroSpot read-out

3 or 4 x $10^5$ cells were seeded per well in 6-well plates and transfected one day after seeding with M^pro plasmids using TransIT-PRO (Mirus Bio LLC) and incubated 8–9 hours for M^pro-On and 10 hours for M^pro-Off assays. Then, cells were seeded into a 96-well plate with 2 x $10^4$ cells per well in 50 μl complete growth medium. Directly after seeding, compounds and virus

(MOI 0.1) were added in 50 µl complete growth medium to wells. After 48 to 72 hours, supernatants were removed, and fluorescent spots counted in a Fluoro/ImmunoSpot counter (CTL Europe GmbH). For experiments where FluoroBrite medium was used, supernatants were not removed, and the signal was read out every 12 hours. The manufacturer-provided software CTL switchboard 2.7.2. was used to scan 90% (70% when FluoroBrite medium was used) of each well area concentrically to exclude reflection from the well edges, and counts were normalized to the full area. Automatic fiber exclusion was applied while scanning. The excitation wavelength for dsRed was 570 nm, and the D_F_R triple band filter was used to collect fluorescence. To increase comparability between M$^{pro}$-On and -Off signals, we normalized dsRed events with the following strategies. In M$^{pro}$-On, the highest compound concentrations would not reach the same value due to the different responses of each mutant. Therefore, we normalized to the mean of the respective parental proteases (WT and Omicron). In M$^{pro}$-Off, we normalized the signal to either the wild type or the Omicron proteases. For both assays, fluorescent spot count (y-axis) data were plotted against PI concentration (x-axis) and IC$_{50}$ values were extrapolated as described later in this section.

## Replication kinetic experiments

For replication kinetic experiments with WT M$^{pro}$, Omicron and mutants, we adapted the 3CL/M$^{pro}$-Off assay to enable multiple read-outs over time every 12 hours. We exchanged the standard DMEM supplemented with 10% FCS, P/S, 2% L-glutamine, 1x sodium pyruvate and 1x non-essential amino acids (Gibco), with FluoroBrite DMEM, equally supplemented. FluoroBrite DMEM has the same composition as DMEM, except it lacks phenol red which interferes with spot recognition. After seeding $2 \times 10^4$ transfected cells in 50 µl of FluoroBrite complete medium into 96-well plates, only virus (MOI 0.1) was added (50 µl). The signal was read out at approximately 12 hours intervals. As the virus starts replicating, dsRed is expressed by the infected cells, and the expression of dsRed is correlated with the amount of viral progeny and therefore with the number of spot count. We generally stopped signal acquisition after 84 up to 108 hours post infection, because the cells die due to virus-induced cytopathy, and the spot count decreases. The manufacturer-provided software CTL switchboard 2.7.2. was used as described above (for FluoroBrite medium). Replication kinetic curves were normalized individually to the highest mean of the construct. We normalized fluorescent spot count data by the highest mean of each dataset. Then, we plotted the fluorescent dsRed signal (spot count) against time (hours). We then performed non-linear regression analyses using the built-in function "Sigmoidal 4PL, X is concentration" in GraphPad Prism 9.5, where concentration was replaced with "hours" and extrapolated the TM$_{50}$ (time-maximum fifty) value, which represents the time required to achieve half of the maximum signal/plateau.

## Protein expression and purification

M$^{pro}$ with a C-terminal Hexahistdine tag (M$^{pro}$-6H) was expressed in *E. coli* followed by a tag removal with the human rhinovirus 14 3C protease (HRV3C protease).

WT M$^{pro}$-6H construct was codon optimized for the expression in *E. coli* and ordered from Thermo Scientific (Waltham, MA, USA). Point mutations to generate the mutant variants were introduced using the Q5 Site-Directed Mutagenesis Kit purchased from New England Biolabs (NEB, Ipswich, MA, USA). Primers were acquired from Sigma Aldrich (St. Louis, MO, USA).

M$^{pro}$-6H variants were transformed into electrocompetent *E. coli* BL21(DE3) cells. An overnight culture was inoculated 1:50 in 1L TB medium and cells were grown at 37˚C, 220 rpm to an OD$_{600nm}$ of 1. After induction with 0.4 mM Isopropyl β-D-1-thiogalactopyranoside (IPTG)

temperature was lowered to 25˚C for 4 hours. Cells were isolated by centrifugation at 3200 g, 20 minutes at 4˚C, and cell pellets frozen at -20˚C.

For purification, frozen pellets were resuspended in 50 mM Tris/50mM NaCl, pH 7.5 buffer and disrupted with a French press (Thermo Scientific). Lysed cells were centrifuged at 20,000 g, 4˚C, 20 min and the clarified supernatant was adjusted to 50 mM Tris/300 mM NaCl/20 mM Imidazole. IMAC purification was carried out with a ÄKTApurifier (GE Healthcare Life Sciences; Little Chalfon, United Kingdom) and a HisTrap FF Crude 1 mL (GE Healthcare) column. Unbound proteins were washed out using 4 column volumes (CV) 50 mM Tris/300 mM NaCl/20 mM Imidazole, pH 7.5 and 1 CV 50 mM Tris/300 mM NaCl/40 mM Imidazole, pH 7.5. The protein-containing fractions were pooled, and the buffer was changed to 50 mM Tris/50mM NaCl, pH 7.5 using a HiTrap Desalting 5 mL (GE Healthcare) column. Protein concentration was determined with a NanoDrop ND-1000 Spectrometer (Thermo Scientific).

### Hexahistidine Tag removal by negative IMAC

The highly soluble Hexahistidine tagged NT*-HRV3Cprotease was expressed as previously described [76] and purified the same way as M^pro. M^pro-6H was incubated together with NT*-HRV3protease at 4˚C overnight followed by negative IMAC purification with a manually packed Ni-NTA column. The flowthrough yields the mature M^pro with native N- and C-termini without His tag, because the His tag as well as the NT*-HRV3C protease are bound to the Ni-NTA resin.

### Cross validation with biochemical M^pro inhibition assay

The biochemical assay used to confirm mutations was based on the 3CL^pro / M^pro activity assay from BPS Biosciences, catalog number #78042–2. The M^pro in the kit was replaced by an in-house produced M^pro and mutants thereof (L57F, L167F7P168S, L167F/P168S/L57F), as described in the M^pro purification paragraph. Solutions of WT and mutant M^pro variants were prepared in appropriate buffer (20 mM Tris/HCl pH = 8, 150 mM NaCl, bovine serum albumin 0.1 mg/ml, 1 mM dithiothreitol–DTT) to reach a final concentration of 250 nM in 50 µl. Solutions of Omicron and Omicron-M^pro variants were prepared in the same buffer, to reach a final concentration of 1 µM. 10 µl of five-fold excess to tested (final) nirmatrelvir or ensitrelvir concentrations (180 µM for WT M^pros and 90 µM for Omicron M^pros) were added to the 30 µl of M^pro solution and incubated for 1 hour. Then, 10 µl of a fluorescent substrate (Ac-Abu-Tle-Leu-Gln-MCA, 5 mM) were added to get a final concentration of 40 µM. This generates a 1:5 dilution of the excess of protease inhibitors–and therefore final concentrations–and the reactions were incubated for 4 hours. Fluorescence was measured by excitation at 365 nm and read-out at 415–445 nm emission with a Glomax Explorer fluorometer (Promega).

### Kinetic characterization of recombinant M^pro variants

Enzymatic reactions *in vitro* were carried out in a buffer containing 20 mM Tris×HCl (pH 8.0), 150 mM NaCl, 100 µM TCEP, 1% (v/v) DMSO and 1 mg/mL bovine serum albumin. Recombinant expressed and purified M^pro Wuhan variants and the Omicron variant were used at 2 µM; the M^pro variants carrying the T198I and/or the A206T mutation were used at 10 µM to compensate for their slower kinetics. The oligopeptide N-TSAVLQ↓SGYRKW-C (termed 12YW, M^pro cleavage site indicated) was used as a substrate mimic of the nsp4/5 junction in a serial dilution up to 1 mM. We used a similar substrate as described by Kao et al. [77], but exchanged F for Y at the P3'-position. Reaction components were incubated with stirring at room temperature for 1 min, the enzymatic reaction was quenched by the addition of acetonitrile and trifluoroacetic acid (TFA) (final concentration 5% (v/v) and 0.1% (v/v),

respectively. Reaction products were separated via reversed phase high pressure liquid chromatography on a RP C18 column using a gradient of 15% to 26% acetonitrile in water, 10% methanol and 0.1% TFA with far-UVC detection (215 nm) and a flow rate of 1.3 mL/min. Samples were injected in duplicates and experiments performed three times each. Quantification of the cleavage products was done via peak integration using chromeleon software (7.2.10) against a standard curve established for a serial dilution of the substrate peptide. Calculation of the Michaelis Menten constant $K_M$ and $V_{max}$ was done using the software Graph-Pad Prism (10.2.0). The turnover number ($k_{cat}$) was calculated via $k_{cat} = V_{max} / Et$, where Et is the total enzyme concentration. The catalytic efficiency was calculated via the following equation: efficiency = $k_{cat} / K_M$.

## Preparation of lysates for western blotting

The cells expressing the $M^{pro}$ cutting reporter and the purified proteins were prepared as described before. 50 µl of the protein suspension (5 µg of protein) were pre-incubated with inhibitor (10 µM) or DMSO for 15 minutes at room temperature (R.T.). After that, 25 µl of cell suspension were added and the mixture was incubated for 3 hours at R.T. The cleavage reaction was stopped by the addition of 5x SDS-Loading buffer. Before the SDS-Page the samples were cooked at 95°C for 10 minutes. Western blot images were obtained using the ChemiDoc MP from Bio-Rad.

## TCID$_{50}$ assay and dose responses

For initial dose response experiments, $5 \times 10^4$ BHK-21 cells per well were seeded in 48-well plates one day before infection. Cells were infected in duplicates with a MOI of 0.05 of VSV-$M^{pro}$ WT or VSV-O-$M^{pro}$ and indicated concentrations of nirmatrelvir were added to the wells. After 48 hours, supernatants were collected and titrated with TCID$_{50}$. For quantification, the 50% tissue culture infective dose (TCID$_{50}$) assay was performed as described previously [78]. In short, 100 µl of serial dilutions of virus were added in octuplicates to $10^3$ BHK-21 cells seeded in a 96-well plate. Six days after infection, the TCID$_{50}$ were read out and titers were calculated according to the Kaerber method [79].

## Construction of SARS-CoV-2 $M^{pro}$ mutant replicons

To introduce $M^{pro}$ mutations to the SARS-CoV-2 replicon, BAC recombineering was performed as previously described [46,80]. First, the pSMARTBac-T7-SARS-CoV-2 replicon was transformed into SW102 *E. coli* along with a fragment containing an ampicillin (Amp) cassette flanked by AscI restriction sites and adjacent homologous sequences of $M^{pro}$, resulting in the creation of pSMARTBac-T7-SARS-CoV-2-$M^{pro}$-Amp. Next, the pSMARTBAC-T7-SARS-CoV-2-replicon-Amp DNA was prepared using NucleoBond BAC 100 kit (Takara). A fragment containing the $M^{pro}$ mutation sequence was synthesized with a 30-bp overlap with the upstream and downstream regions of the $M^{pro}$ sequence for assembly. Following AscI (NEB) digestion, the pSMARTBAC-T7-SARS-CoV-2-replicon-AMP DNA was ligated to the fragment containing the $M^{pro}$ mutations using Gibson Assembly. This resulted in the generation of pSMARTBAC-T7-SARS-CoV-2-$M^{pro}$ mutants replicons.

## Protease inhibition assay using SARS-CoV-2 replicon mutants

SARS-CoV-2 replicons RNA were prepared as previously described [46]. Briefly, $1 \times 10^6$ Huh-7 cells were electroporated with 2 µg of replicon RNA and subsequently plated into 96-well plates (Corning, cat. 3904) with a seeding density of $1 \times 10^4$/well. Immediately after

electroporation, a four-fold serial dilution of protease inhibitors (nirmatrelvir or ensitrelvir) was added to the wells. DMSO only was added to the control wells and served as solvent control. At 48 hours post-transfection, the reporter activities of in the cells treated with inhibitors were measured by quantifying the number of green cells in each well using an Acumen eX3 scanner and normalized it to the number of green cells in DMSO only wells (control wells).

### Pearson's correlation coefficient calculation of M$^{pro}$-Off and SARS-CoV-2 replicon data

A simple R script (R version 4.2.2) was used to calculate the Pearson's correlation coefficients for the whole datasets. The datasets comprise $IC_{50}$ fold change values (vs. WT) obtained from M$^{pro}$-Off and SARS-CoV-2 replicon experiments. The correlations were performed for nirmatrelvir and ensitrelvir altogether, and nirmatrelvir or ensitrelvir only). The calculation of the correlation coefficient was performed by using the built-in function ´cor()´ in R, taking as arguments the two datasets to be correlated and the correlation method = "pearson".

### $IC_{50}$ (and $TM_{50}$) calculations

In this study, different assay systems were used to generate resistance data, namely VSV-based cellular assays with FluoroSpot read out, a luminescence-based cellular assay that employs recombinant M$^{pro}$, as well as a peptide-MCA cleavage-based biochemical assay and SARS-CoV-2 mutant replicons assays. $IC_{50}$ calculations and statistical analysis for all assays were performed with GraphPad Prism 10. To calculate $IC_{50}$s, we used GraphPad's pre-set sigmoidal models: 4PL, X is concentration (hours in case of the replication kinetics).

$$Y = Bottom + \frac{Top - Bottom}{1 + \left(\frac{IC_{50}}{X}\right)^{HillSlope}}. \qquad \text{Eq 1}$$

However, under some circumstances, slightly bell-shaped curves arose. In the On assay, the decrease occurs at the end because of inhibitor toxicity at high concentrations. In the Off assay, the initial decrease can be attributed to the fast death of cells in the absence of any inhibitor, whereas cells die slower at small doses, allowing for more cell division and thereby more "substrate" (cells and virus) generating dsRed. We chose to compensate for such unrealistically steep slopes by constraining the HillSlope to 3 or lower (positive slope) for the gain-of-signal assay. In dose response M$^{pro}$-Off experiments, we did not apply a constraint in the HillSlope parameter, since it did not change $IC_{50}$ values meaningfully. However, in M$^{pro}$-Off kinetic experiments we observed steep signal increases more frequently. Therefore, the fitting was performed by setting a constraint in the HillSlope ($<$ 15), allowing for more precise determination of the confidence intervals.

### Hardware overview (TTMD simulations)

Most of the molecular modeling operations, such as the structure preparation, the setup for molecular dynamics (MD) simulations, and trajectory analyses, were performed on a Linux workstation, equipped with a 20 cores Intel Core i9-9820X 3.3 GHz processor, running the Ubuntu 20.04 operating system. Molecular dynamics simulations were carried out on an in-house Linux GPU cluster composed of 20 NVIDIA devices ranging from GTX1080Ti to RTX3090.

### Structure preparation

A computational study was conducted to rationalize the effect of SARS-CoV-2 M$^{pro}$ mutations on nirmatrelvir and ensitrelvir resistance. In detail, a recently developed enhanced sampling

molecular dynamics method named Thermal Titration Molecular Dynamics (TTMD) [58,81] was exploited to compare the stability of the non-covalent complex between nirmatrelvir and both the WT and mutated proteases, based on the assumption that the formation of the reversible, non-covalent complex, is the rate-determining step that leads to the formation of the irreversible (or very slowly reversible), covalent complex [82]. The top five mutants concerning their experimentally determined drug resistance were investigated. Similarly, TTMD was also applied to the non-covalent complex between ensitrelvir and both the WT and mutated proteases, investigating three mutants.

The experimentally determined three-dimensional coordinates of the WT nirmatrelvir-M^pro and ensitrelvir-M^pro complexes were retrieved from the Protein Data Bank (PDB) [83] and prepared for further calculations exploiting various tools provided within the Molecular Operating Environment (MOE) 2022.02 suite [84]. Particularly, the complexes deposited with accession codes 7RFW [6] and 8DZ0 [51] were used for describing the WT protease, while mutants were generated through the "protein builder" module of MOE.

The catalytically competent dimeric conformation of the protease was reconstructed by applying symmetric crystallographic transformations. Inconsistencies in the experimental structure were checked and fixed through the "structure preparation" tool. Missing hydrogen atoms were added according to the predominant tautomeric and protomeric state of each residue at pH = 7.4 through the "Protonate3D" module. Finally, each water molecule was removed before storing the prepared complex structure for subsequent calculations. In the end, 1:1 protein-ligand non-covalent complexes between nirmatrelvir/ensitrelvir and various M^pro proteases were considered in the simulated systems.

## System setup for MD simulations and Equilibration Protocol

Each complex obtained in the previous step was prepared for MD simulation making use of various tools provided within Visual Molecular Dynamics 1.9.2 (VMD) [85] and the Amber-Tools 22 [86,87] suite. All parameters were attributed according to the ff14SB [88] force field, except for the ligand ones, which were instead assigned through the general Amber force field (GAFF) [89]. Partial charges for the ligand were calculated with the AM1-BCC method [90]. Each one of the protein-ligand systems was solvated using the TIP3P [91] water model into a rectangular base box, ensuring a minimum 15 Å distance between the box border and the nearest protein or ligand atom. To neutralize the net charge of the system and to reach a salt concentration of 0.154 M, the proper number of sodium and chlorine ions was added. Finally, to remove clashes and unfavorable contacts, 500 steps of energy minimization through the conjugate gradient algorithm were performed.

Before the production phase, a two-step equilibration process was performed. First, 0.1 ns of simulation in the canonical ensemble were carried out, imposing harmonic positional restraints on both the protein and ligand atoms. Then, 0.5 ns of simulation in the isothermal-isobaric ensemble were performed, with harmonic positional restraints applied only on the protein backbone and ligand atoms. In each case, a 5 kcal mol$^{-1}$ Å$^{-2}$ force constant was imposed on each restrained atom for the whole duration of the simulation, and the temperature was maintained at a constant value of 300 K through a Langevin thermostat [92]. In the second equilibration stage, the pressure was kept constant at 1 atm through a Monte Carlo barostat [93].

All MD simulations were conducted exploiting the ACEMD16 3.5 engine, which is based upon the open-source Python library for biomolecular simulations OpenMM [94]. A 2 fs integration timestep and the M-SHAKE algorithm were applied to constrain the length of bonds involving hydrogen atoms. The particle-mesh Ewald method was used to compute electrostatic

interactions, using cubic spline interpolation and a 1 Å grid spacing. A 9.0 Å cutoff was applied to the computation of Lennard-Jones interactions.

## Thermal Titration Molecular Dynamics (TTMD) simulations

As thoroughly described in the work of Pavan et al. [58], Thermal Titration Molecular Dynamics (TTMD) is an enhanced sampling MD protocol originally developed for the qualitative estimation of protein-ligand unbinding kinetics. This approach allows the characterization in high detail of the binding mode and residence time of a ligand within the catalytic site of a given protein-ligand complex. It thereby enables the comparison between: 1. different ligands bound on the same protein [58]; 2. different ligand conformations within the same binding site; 3. different mutants of the same protein in complex with the same ligand, as in the present work. The simulation consists of a series of classic molecular dynamics simulations performed at progressively increasing temperatures. Whether the ligand remains bound or not, is monitored through a scoring function based on protein-ligand interaction fingerprints (IFP$_{CS}$) [95]. This simulation returns a score named MS coefficient, that indicates the stability/instability of the inhibitor-protein complex. The MS coefficient average calculation was performed according to the protocol described in its original publication [58]: five independent TTMD simulations were carried out for each generated complex, and the MS score was averaged across three replicates, discarding the highest and lowest values.

The TTMD workflow relies on a series of short classic molecular dynamics simulations, defined as "TTMD-steps", performed at progressively increasing temperatures. The temperature increase is used to augment the kinetic energy of the system, thus shortening the simulation time required to observe protein-ligand unbinding events compared to classic MD simulations. To monitor the progress of the simulation, the conservation of the native binding mode is evaluated through a protein-ligand interaction fingerprint (PLIF) based scoring function [95]. The protocol described hereafter is implemented as a Python 3.10 code relying on the NumPy, MDAnalysis [96,97], Open Drug Discovery Toolkit (ODDT) [98] and Scikit-learn packages. The code is released under a permissive MIT license and available free of charge at github.com/molecularmodelingsection/TTMD.

In detail, the user must define a "temperature ramp", i.e., the number, the temperature, and the length of each "TTMD-step". Consistently with previous works on the target [58,81], in this case, the starting and end temperatures were set at 300K and 450K respectively, with a temperature increase between each "TTMD-step" of 10K and the duration of each simulation window set at 10 ns. The extension of the temperature ramp was determined based on the conservation of the native fold of the protein throughout the simulation, monitored through the backbone RMSD.

## Structure preparation (analyses with Bioluminate and Osprey)

The PDB entries 7ALI [56], 8DZ0 [51], 8DZ2 [51], 7DVW [52], 7TLL [57] and 8HOL [99] were used to investigate the impact of mutations on apo WT, ensitrelvir-bound WT, nirmatrelvir-bound WT, nsp5/6-bound H41A mutant, nirmatrelvir-bound and apo Omicron dimers. All structures were prepared with the default settings of the Protein Preparation Wizard in Maestro version 2022–3 [100] except that seleno-methionines were converted to methionines and no water molecules were deleted. Only the structure 7ALI [56] and 7DVW [52] includes residue 305 in both chains, 8DZ0 [51] do and 8DZ2 [51] only includes residue 305 in chain B. Structure visualization and figure creation were performed in PyMol [101] version 2.5.0.

## Stability prediction

The stability of all the mutations was calculated starting from the WT or Omicron M^pro structures using the default settings of the residue_scanning_backend.py module in Bioluminate [47–50] release 2022–3. Mutations were introduced in combination in the two protomers.

## Affinity prediction

For the affinity predictions in Osprey version 3.3 [54,102–104] water molecules were not considered. Osprey calculates so called Log10 K* scores [105] which provide an estimation of binding affinity. The required yaml and frcmod input files were created as described in detail in the Guerin et al. STAR Protocol [105]. For the dimer affinity calculations, chain B was considered as the ligand. The stability threshold was disabled, epsilon was defined as 0.03 and WT and mutant side chain conformations were calculated as continuously flexible. Osprey is an open-source software and is available for free at https://github.com/donaldlab/OSPREY3.

## Supporting information

**S1 Fig. VSV-M^pro and VSV-Omicron-M^pro are equally susceptible to nirmatrelvir.** (**A**) Schematic representation of VSV-M^pro and VSV-Omicron-M^pro genomes. VSV-Omicron-M^pro was generated by introducing the substitution P to H at amino acid position 132 (P132H). (**B**) Dose response curves of VSV-M^pro and VSV-Omicron-M^pro against nirmatrelvir. Data are presented as geometric mean of n = 2 biologically independent replicates per condition. Each biological replicate consisted of n = 8 technical replicates. (**C**) Crystal violet staining of BHK21 cells used for dose response experiments with VSV-M^pro. Data are presented as the mean of n = 3 biologically independent replicates per condition.
(TIF)

**S2 Fig. Different parental proteases and related mutations after selection experiments.** (**A**) Schematic representation of the selection experiments. Mutant pools are described in more detail in S1 Table. (**B**) Schematic representation of the different parental proteases selected and used for further selection experiments with nirmatrelvir. From top to bottom: L167F-M^pro, L167F/P168S-M^pro, L167F/F305L-M^pro, P132H-M^pro (Omicron-M^pro) and O/A206T-M^pro. Mutations are represented by column-like symbols along the proteases, according to their position in the M^pro sequence.
(TIF)

**S3 Fig. Muscle alignment of M^pro and related coronavirus proteases.** MUltiple Sequence Comparison by Log-Expectation (MUSCLE) sequence alignment of SARS-CoV-2, SARS-CoV-1, Bat-CoV HKU9, PEDV, MHV (Mouse hepatitis virus), MERS, IBV, NL63 and 229E shows areas of conservation and amino acid variability.
(TIF)

**S4 Fig. Phylogenetic subtrees of T198I, P184S, K100N, and F8L substitutions.** (**A**) Phylogenetic subtree of nsp5/M^pro-T198I, -P184S, -K100N, -F8L substitutions generated with the Ultrafast Sample placement on Existing tRee (UShER) tool (GISAID, 18th January 2023). Only sequences deposited after the Omicron emergence were used. (**B**) Phylogenetic subtree of nsp5/M^pro-T198I and magnified view of the T21I subtree areas, showing transmission of this variant from a single founder event. (**C**) Frequency ratio of T198I, P184S, K100N, and F8L mutations compared with pre- and post-Omicron variant surge (GISAID, 18^th of January 2023).
(TIF)

**S5 Fig. Comparison of the cell-based systems used in this study.** From left to right, schematic representation of: construct(s), mechanism in the absence of inhibitor, mechanism in the presence of inhibitor. (**A**) VSV-M$^{pro}$ (mutation selection tool) construct and mechanism. Without inhibitor, M$^{pro}$ processes G-M$^{pro}$-L and the virus replicates. After an inhibitor is applied, M$^{pro}$ is inactive and viral replication is blocked. (**B**) Graphical representation of the loss-of-signal cellular assay mechanism: VSV-ΔL-dsRed + G-M$^{pro}$-L are added to cells. Without inhibitor applied, M$^{pro}$ processes GFP-M$^{pro}$-L and VSV-ΔL-dsRed replicates, expressing dsRed. After an inhibitor is applied, M$^{pro}$ is inactive and VSV-ΔL-dsRed replication is turned off. (**C**) Graphical representation of the gain-of-signal cellular assay mechanism: VSV-ΔP-dsRed + P:M$^{pro}$:P are added to cells. Without inhibitor, M$^{pro}$ cleaves the phosphoprotein (P) in two pieces, impairing viral replication of replication-incompetent VSV-ΔP-dsRed. After an inhibitor is applied, M$^{pro}$ is inactive, P is intact and VSV-ΔP-dsRed replication is turned on and expresses dsRed.
(TIF)

**S6 Fig. M$^{pro}$-On dose response curve of WT and Omicron mutants main proteases.** (**A**) Gain-of-signal assay results are shown for M$^{pro}$ WT and L167F, F305L, F8L, L57F, P184S, P168S mutants against the protease inhibitors nirmatrelvir (top) and ensitrelvir (bottom). Representative experiment is shown. Data is presented as mean ± SEM of 2 biologically independent replicates per condition. (**B**) Gain-of-signal assay results are shown for M$^{pro}$ WT and S144A, L167F/F305L, L167F/P168S, L167F/F305L/S144A, L167F/F305L/P184S, L167F/F305L/F8L, L167F/F305L/L57F mutants against the protease inhibitors nirmatrelvir (top) and ensitrelvir (bottom). Representative experiment is shown. Data is presented as mean ± SEM of 2 biologically independent replicates per condition. (**C**) Gain-of-signal assay results are shown for M$^{pro}$ WT and T21I, L167F/F305L/P184S/T21I mutants against the protease inhibitors nirmatrelvir (top) and ensitrelvir (bottom). Representative experiment is shown. Data is presented as mean ± SEM of 2 / 3 biologically independent replicates per condition. (**D**) Gain-of-signal assay results are shown for Omicron-M$^{pro}$ and O/A266T, O/A210S, O/Y54H, O/Y54H +F305L, O/R188W, O/R222L, O/R188W/R222L mutants against the protease inhibitor nirmatrelvir (top) and ensitrelvir (bottom). Representative experiment is shown. Data is presented as mean ± SEM of 2 biologically independent replicates per condition. (**E**) Gain-of-signal assay results are shown for Omicron-M$^{pro}$ and O/A206T, O/T198I, O/K100N, O/A206T/F8L, O/A206T/K100N mutants against the protease inhibitor nirmatrelvir (top) and ensitrelvir (bottom). Representative experiment is shown. Data is presented as mean ± SEM of 3 biologically independent replicates per condition.
(TIF)

**S7 Fig. M$^{pro}$-Off dose response curve of WT and Omicron mutants main proteases.** (**A**) Representative loss-of-signal assay results at 48 hours post infection (hpi) are shown for M$^{pro}$ WT and F8L, L57F, T21I, P184S, S144A mutants against the protease inhibitors nirmatrelvir (top) and ensitrelvir (bottom). Data is presented as SEM of n = 3/n = 4 biologically independent replicates per condition. Fold-changes were calculated using the IC$_{50}$ value of WT M$^{pro}$ at this timepoint. (**B**) Representative loss-of-signal assay results at 60 hpi are shown for M$^{pro}$ WT and L167F, L167F/F305L, L167F/P168S/L57F, L167F/F305L/P184S/T21I mutants against the protease inhibitors nirmatrelvir (top) and ensitrelvir (bottom). Data is presented as SEM of n = 3/n = 4 biologically independent replicates per condition. Fold-changes were calculated using the IC$_{50}$ value of WT M$^{pro}$ at this timepoint. (**C**) Representative loss-of-signal assay results at 72 hpi are shown for M$^{pro}$ WT, L167F/F305L/F8L and L167F/F305L/P184S mutant against the protease inhibitors nirmatrelvir (top) and ensitrelvir (bottom). Data is presented as SEM of n = 3/n = 4 biologically independent replicates per condition. Fold-changes were

calculated using the IC$_{50}$ value of WT M$^{pro}$ at this timepoint. (**D**) Representative loss-of-signal results at 84 hpi are shown for M$^{pro}$ WT and L167F/F305L/S144A mutants against the protease inhibitors nirmatrelvir (top) and ensitrelvir (bottom). Data is presented as SEM of n = 3/n = 4 biologically independent replicates per condition. Fold-changes were calculated using the IC$_{50}$ value of WT M$^{pro}$ at this timepoint. (**E**) Representative loss-of-signal assay results at 48 hpi are shown for Omicron-M$^{pro}$ and O/A266T, O/A210S, O/Y54H, O/K100N, O/Y54H+F305L mutants against the protease inhibitor nirmatrelvir (top) and ensitrelvir (bottom). Data is presented as SEM of n = 3/n = 4 biologically independent replicates per condition. Fold-changes were calculated using the IC$_{50}$ value of Omicron M$^{pro}$ at this timepoint. (**F**) Representative loss-of-signal assay results at 48 hpi are shown for Omicron-M$^{pro}$ and O/R188W, O/R222L, O/R188W/R222L mutants against the protease inhibitor nirmatrelvir (top) and ensitrelvir (bottom). Data is presented as SEM of n = 4 biologically independent replicates per condition. Fold-changes were calculated using the IC$_{50}$ value of Omicron M$^{pro}$ at this timepoint. (**G**) Representative loss-of-signal signal assay results at 60 hpi are shown for Omicron-M$^{pro}$ and O/ A206T, O/T198I, O/A206T/T198I, O/A206T/K100N mutants against the protease inhibitor nirmatrelvir (top) and ensitrelvir (bottom). Data is presented as SEM of n = 3/n = 4 biologically independent replicates per condition. Fold-change was calculated using the IC$_{50}$ value of Omicron M$^{pro}$ at this timepoint. (**H**) Representative loss-of-signal signal assay results at 60 hpi are shown for Omicron-M$^{pro}$ and O/A206T, O/A206T/F8L mutants against the protease inhibitor nirmatrelvir (top) and ensitrelvir (bottom). Data is presented as SEM of n = 3/n = 4 biologically independent replicates per condition. Fold-change was calculated using the IC$_{50}$ value of Omicron M$^{pro}$ at this timepoint.
(TIF)

**S8 Fig. IC$_{50}$ fold changes of M$^{pro}$-Off WT & Omicron mutants.** (**A**) Loss-of-signal assay results are shown for all the WT M$^{pro}$ mutants. Data is presented as mean and SD of n = 2 / n = 3 or n = 4 independent experiments. (**B**) Loss-of-signal assay results are shown for all the Omicron M$^{pro}$ mutants. Data is presented as mean and SD of n = 2 / n = 3 independent experiments.
(TIF)

**S9 Fig. M$^{pro}$-Off kinetics mechanism & normal DMEM/FluoroBrite DMEM culture media differences.** (**A**) Schematic representation of the M$^{pro}$-Off assay adaptation to a viral replication kinetic measurement based on M$^{pro}$ activity. M$^{pro}$-Off transfected cells are infected with VSV-ΔL-dsRed and the signal is read out over time. No protease inhibitor is applied. (**B**) Representative photo, and magnified view (below), taken with the ELISpot reader (FluoroSpot X suite) of a well after removing the supernatant. (**C**) Representative photo, and magnified view (below), taken with the ELISpot reader (FluoroSpot X suite) of a well where the supernatant was not removed (DMEM, + phenol red). (**D**) Representative photo, and magnified view (below), taken with the ELISpot reader (FluoroSpot X suite) of a well where the supernatant was not removed (FluoroBrite DMEM,—phenol red). (**E**) Representative replication kinetics fitting curves of WT M$^{pro}$ and mutants F8L, T21I, P184S, L57F, S144A, L167F, L167F/P168S, L167F/P168S/L57F, L167F/F305L, L167F/F305L/F8L, L167F/F305L/S144A, L167F/F305L/ P184S, L167F/F305L/P184S/T21I (± SD; n = 12 biologically independent replicates). The dotted line represents the TM$_{50}$ value related to the WT M$^{pro}$. (**G**) Replication kinetics fitting curves of Omicron, O/Y54H, O/Y54H/F305L, O/A210S, O/A266T, O/K100N M$^{pro}$ mutants (± SD; n = 8 biologically independent replicates). The dotted lines represent the TM$_{50}$ value related to the Omicron main protease and the mutant O/A210S. (**H**) Replication kinetics fitting curve of Omicron, O/R188W, O/R222L, O/R188W/R222L M$^{pro}$ mutants (± SEM; n = 8 biologically independent replicates). (**I**) Replication kinetics fitting curves of Omicron, O/

A206T, O/A206T/F8L, O/A206T/T198I, O/A206T/K100N, O/T198I M$^{pro}$ mutants (± SEM; n = 8 biologically independent replicates). The dotted lines represent the TM$_{50}$ value related to the Omicron main protease and the mutant O/A206T/F8L. (TIF)

**S10 Fig. Selected recombinant proteases.** (**A**) Following recombinant protein expression, M$^{pro}$ variants were purified using a FPLC system for subsequent assays employing the recombinant protease. (**B**) Coomassie stained denaturing protein gel detailing purification steps of M$^{pro}$. (**C**) Scheme of the M$^{pro}$-cutting reporter. The cutting sequence of M$^{pro}$ is flanked by two fragments of the reporter protein. Incubation of the reporter with purified M$^{pro}$ leads to reporter cleavage. Protease inhibitor binding and mutations can stop the cleavage event. (**D**) Proteolytic activity assessment via western blot of uncleaved reporter protein compared to WT Wuhan-1 M$^{pro}$ and mutants thereof. (**E**) Fluorogenic substrate peptide coupled with MCA, which releases fluorescent AMC upon cleavage by M$^{pro}$. Dose response experiments of WT and C145A mutant (catalytically inactive) were performed with nirmatrelvir and ensitrelvir (± SEM; n = 2 technical replicates). (**F**) Schematic representation of the HPLC-based peptide cleavage assay. Magnified, a representative chromatogram showing the cleaved peptide (green text) and the uncleaved peptide (red text) peaks. (TIF)

**S11 Fig. Biochemical/enzymatic assays, computational stability and dimerization affinity data.** (**A**) Left: dose response experiment of WT, L57F, L167F/P168S and L167F/P168S/L57F with nirmatrelvir (± SD; n = 2 technical replicates). Right: dose response experiment of WT, L57F, L167F/P168S and L167F/P168S/L57F with ensitrelvir (± SD; n = 2 technical replicates). (**B**) Left: dose response experiment of Omicron, O/A206T and O/A206T/T198I with nirmatrelvir (± SD; n = 2 technical replicates). Right: dose response experiment of Omicron, O/A206T and O/A206T/T198I with nirmatrelvir (± SD; n = 2 technical replicates). (**C**) Michaelis-Menten curves of WT, L57F, L167F/P168S and L167F/P168S/L57F proteases (± SD, n = 3 independent experiments). All the proteases were tested at 2 μM. (**D**) Michaelis-Menten curves of Omicron, A206T, A206T/T198I proteases (± SD, n = 3 independent experiments). Omicron protease was tested at 2 μM, whereas O/A206T and O/A206T/T198I were tested at 10 μM. (**E**) Plot of Δ_Stability values for mutations introduced to the nsp5/6-M$^{pro}$ structure (PDB entry 7DVW) and in the nirmatrelvir bound structure (PDB entry 8DZ2). Inside the plot, a magnified subplot highlighting some data points. Some of the mutations that were investigated in this work are labelled here. (**F**) Dimerization affinity plot: positive values mean increase in dimerization affinity, negative values indicate decreased or impaired affinity. Each data point represents a mutation, which is colored based on the parental protease they have arose from (i.e. F8L is colored in light sea green and it represent the triple mutant L167F/F305L/F8L). (TIF)

**S12 Fig. Thermal Titration Molecular Dynamics (TTMD) experiments of nirmatrelvir in complex with WT, L167F/F305L/S144A, L167F/F305L/P184S and L167F/F305L/F8L.** (**A**) Simulation with the WT protease. Left: rainbow plot and Root Mean Square Deviation (RMSD) of backbone and ligand. Middle: structural overlay of pre- and post-TTMD simulation. Right: titration profile, from which the MS coefficient can be extrapolated. (**B**) Simulation with the L167F/F305L/S144A mutant protease. Top: rainbow plot and Root Mean Square Deviation (RMSD) of backbone and ligand. Middle: structural overlay of pre- and post-TTMD simulation. Bottom: heat map with interaction energies between the ligand and the surrounding residues. (**C**) Simulation with the L167F/F305L/P184S mutant protease. Top: rainbow plot and Root Mean Square Deviation (RMSD) of backbone and ligand. Middle: structural overlay

of pre- and post-TTMD simulation. Bottom: heat map with interaction energies between the ligand and the surrounding residues. (**D**) Simulation with the L167F/F305L/F8L mutant protease. Top: rainbow plot and Root Mean Square Deviation (RMSD) of backbone and ligand. Middle: structural overlay of pre- and post-TTMD simulation. Bottom: heat map with interaction energies between the ligand and the surrounding residues.
(TIF)

**S13 Fig.** TTMD simulation data of the ensitrelvir-WT-M^pro (**A**), ensitrelvir- L167F/P168S/ L57F -M^pro (**B**), ensitrelvir-L167F/F305L/P184S-M^pro (**C**) ensitrelvir-L167F/F305L/ P184S-M^pro (**D**) complexes.
(TIF)

**S1 Table. VSV-M^pro mutants generated during selection experiments with the indication of at which passage and concentration they have been selected.**
(DOCX)

**S2 Table. Break-down of mutations with counts >500 in GISAID according to their appearance in a variant of concern.**
(DOCX)

**S3 Table. Cloning primers.**
(DOCX)

**S4 Table. Retrieved mutations from selection experiments and Nanopore sequencing coverage in percentage (%).** Table displaying all the retrieved mutations from different VSV-M^pro variants used: VSV-O/A206T-M^pro, VSV-Omicron-M^pro, VSV-L167F-M^pro, VSV-L167F/ P168S-M^pro and VSV-L167F/F305L-M^pro. For each substitution, the coverage (in percentage) is reported on its right.
(DOCX)

**S1 Data. Minimal dataset.**
(XLSX)

**S1 Movie. TTMD simulations of L167F/P168S/L57F (left) and WT (right) nirmatrelvir complexes.** Top: 3D structures before and after the simulation are colored in light blue and orange, respectively; Bottom: rainbow plot and root-mean-square-deviation plot of the investigated systems.
(MP4)

**S2 Movie. TTMD simulations of L167F/F305L/S144A (left) and WT (right) nirmatrelvir complexes.** Top: 3D structures before and after the simulation are colored in light blue and orange, respectively; Bottom: rainbow plot and root-mean-square-deviation plot of the investigated systems.
(MP4)

**S3 Movie. TTMD simulations of L167F/F305L/P184S (left) and WT (right) nirmatrelvir complexes.** Top: 3D structures before and after the simulation are colored in light blue and orange, respectively; Bottom: rainbow plot and root-mean-square-deviation plot of the investigated systems.
(MP4)

**S4 Movie. TTMD simulations of L167F/F305L/F8L (left) and WT (right) nirmatrelvir complexes.** Top: 3D structures before and after the simulation are colored in light blue and orange, respectively; Bottom: rainbow plot and root-mean-square-deviation plot of the

investigated systems.
(MP4)

**S5 Movie. TTMD simulations of L167F/F305L/P184S/T21I (left) and WT (right) nirmatrelvir complexes.** Top: 3D structures before and after the simulation are colored in light blue and orange, respectively; Bottom: rainbow plot and root-mean-square-deviation plot of the investigated systems.
(MP4)

**S6 Movie. MD simulation of L167F/F305L/F8L "wiper" mechanism.** The C-terminal is colored in dark grey and its movement is shown throughout the simulation.
(MP4)

## Acknowledgments

We thank Florian Krammer, Bernhard Rupp and Sebastiaan Werten for useful discussion, Jasmin Behrens for support in recombinant protein production and Catherine Gardener for spell-checking the manuscript. The MMS lab is grateful to Chemical Computing Group, OpenEye, and Acellera for the scientific and technical partnership. MMS lab thankfully acknowledges the support of NVIDIA Corporation with the donation of the Titan V GPU used for this research.

## Author Contributions

**Conceptualization:** Seyed Arad Moghadasi, Theresia Dunzendorfer-Matt, Emmanuel Heilmann.

**Data curation:** Francesco Costacurta, Andrea Dodaro, David Bante, Helge Schöppe, Ju-Yi Peng, Bernhard Sprenger, Xi He, Seyed Arad Moghadasi, Lisa Maria Egger, Jakob Fleischmann, Matteo Pavan, Davide Bassani, Silvia Menin, Stefanie Rauch, Laura Krismer, Anna Sauerwein, Anne Heberle, Toni Rabensteiner, Theresia Dunzendorfer-Matt, Emmanuel Heilmann.

**Formal analysis:** Francesco Costacurta, Andrea Dodaro, David Bante, Helge Schöppe, Ju-Yi Peng, Bernhard Sprenger, Xi He, Seyed Arad Moghadasi, Lisa Maria Egger, Jakob Fleischmann, Matteo Pavan, Davide Bassani, Silvia Menin, Laura Krismer, Toni Rabensteiner, Theresia Dunzendorfer-Matt, Emmanuel Heilmann.

**Funding acquisition:** Reuben S. Harris, Eduard Stefan, Rainer Schneider, Theresia Dunzendorfer-Matt, Dai Wang, Teresa Kaserer, Stefano Moro, Dorothee von Laer, Emmanuel Heilmann.

**Investigation:** Francesco Costacurta, Andrea Dodaro, David Bante, Helge Schöppe, Ju-Yi Peng, Bernhard Sprenger, Xi He, Lisa Maria Egger, Jakob Fleischmann, Matteo Pavan, Davide Bassani, Silvia Menin, Stefanie Rauch, Anna Sauerwein, Joses Ho, Theresia Dunzendorfer-Matt, Emmanuel Heilmann.

**Methodology:** Francesco Costacurta, Andrea Dodaro, David Bante, Helge Schöppe, Ju-Yi Peng, Bernhard Sprenger, Seyed Arad Moghadasi, Lisa Maria Egger, Jakob Fleischmann, Davide Bassani, Silvia Menin, Anna Sauerwein, Anne Heberle, Toni Rabensteiner, Joses Ho, Theresia Dunzendorfer-Matt, Emmanuel Heilmann.

**Project administration:** Theresia Dunzendorfer-Matt, Dai Wang, Emmanuel Heilmann.

**Resources:** Reuben S. Harris, Eduard Stefan, Rainer Schneider, Theresia Dunzendorfer-Matt, Andreas Naschberger, Dai Wang, Teresa Kaserer, Stefano Moro, Dorothee von Laer, Emmanuel Heilmann.

**Software:** Andrea Dodaro, David Bante, Matteo Pavan, Davide Bassani, Silvia Menin.

**Supervision:** Reuben S. Harris, Eduard Stefan, Rainer Schneider, Theresia Dunzendorfer-Matt, Andreas Naschberger, Dai Wang, Teresa Kaserer, Stefano Moro, Dorothee von Laer, Emmanuel Heilmann.

**Validation:** Francesco Costacurta, Andrea Dodaro, David Bante, Helge Schöppe, Bernhard Sprenger, Seyed Arad Moghadasi, Jakob Fleischmann, Matteo Pavan, Davide Bassani, Silvia Menin, Stefanie Rauch, Laura Krismer, Theresia Dunzendorfer-Matt, Stefano Moro, Emmanuel Heilmann.

**Visualization:** Francesco Costacurta, Andrea Dodaro, David Bante, Helge Schöppe, Bernhard Sprenger, Seyed Arad Moghadasi, Matteo Pavan, Davide Bassani, Emmanuel Heilmann.

**Writing – original draft:** Francesco Costacurta, Emmanuel Heilmann.

**Writing – review & editing:** Francesco Costacurta, Ju-Yi Peng, Theresia Dunzendorfer-Matt, Emmanuel Heilmann.

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
