## [Decision Letter · Decision Letter 0]

27 Dec 2023

Dear Heilmann,

Thank you very much for submitting your manuscript "A comprehensive study of SARS-CoV-2 main protease (Mpro) inhibitor-resistant mutants selected in a VSV-based system" for consideration at PLOS Pathogens. As with all papers reviewed by the journal, your manuscript was reviewed by members of the editorial board and by several independent reviewers.  In light of the reviews (below this email), we would like to invite the resubmission of a significantly-revised version that takes into account the reviewers' comments.

As you can see from their comments, all three reviewers felt that your studies were important, but Reviewers 1 and 2 did raise a number of important points that require your attention. Addressing these points is likely to require both the inclusion of additional experimental data and/or editorial modifications/clarifications within the manuscript.  Please note that with regards to Reviewer 1's major point regarding the need to confirm the impact of the mutations in the context of CoV, should you choose not to perform additional experiments, you should consider expanding your discussion to further address this important point. 

We cannot make any decision about publication until we have seen the revised manuscript and your response to the reviewers' comments. Your revised manuscript is also likely to be sent to reviewers for further evaluation.

Sincerely,

Mark T. Heise

Academic Editor

PLOS Pathogens

Meike Dittmann

Section Editor

PLOS Pathogens

Kasturi Haldar

Editor-in-Chief

PLOS Pathogens

orcid.org/0000-0001-5065-158X

Michael Malim

Editor-in-Chief

PLOS Pathogens

orcid.org/0000-0002-7699-2064

Reviewer's Responses to Questions

**Part I - Summary**

Reviewer #1: In this work, the authors use a previously described VSV system to identify resistance mutations that may arise in SARS-CoV-2 upon passage in protease inhibitors. They propose that the use of this system may avoid the gain-of-function concerns associated with performing these types of studies directly in SARS-CoV-2. Compared to the previous work, this work was done with an Omicron NSP5 and an already partially resistant L167F mutant. Further, some additional variants with a number of mutations were selected for further passaging. They then test the effects of the identified mutations in a variety of assays including VSV replication assays as well as biochemical and in cell protease assays. The also perform a number of computational experiments to understand why these variants likely affect protease activity.

Reviewer #2: In this manuscript, the authors elaborate on a Vesicular Stomatitis Virus (VSV) model they previously established to investigate the evolution of Mpro mutations resistant to protease inhibitors. Given that Mpro is a key antiviral target, the study of drug resistance mechanisms remains to be an important topic, particularly as SARS-CoV-2 infections persist post-pandemic. The manuscript highlights a congruence (although there are clear differences) between mutations identified in their system and those observed in in vitro studies of SARS-CoV-2, suggesting that their BSL-2 model could provide valuable insights into drug resistance pathways with reduced biohazard risks. Utilizing two initial strains, one expressing the Wuhan Mpro with the L167F drug resistant mutation and the other expressing Omicron Mpro with the P132H mutation, the authors document a distinct mutation trajectory conferring resistance, dependent on the initial Mpro variant. They report variations in resistance levels, with some mutations exhibiting heightened resistance to ensitrelvir compared to nirmatrelvir. To explore the underlying mechanisms, the authors employ computational modeling techniques, including Bioluminate and thermal titration molecular dynamics, to predict alterations in the structural stability of Mpro-nirmatrelvir complexes. The manuscript is well-written, with solid experimental design and most conclusions are well-supported. However, it lacks a thorough mechanistic characterization of the Mpro mutants (e.g. missing standard enzyme kinetics) and does not fully explore the reasons behind the differing mutation spectrum arising from the Omicron Mpro versus the Wuhan Mpro. The authors also leave out important information about how nirmatrelvir differs from ensitrelvir and how these differences may lead to unique resistance mechanisms. These areas require further elaboration prior to publication as they are, in this reviewer’s opinion, the significant advances this work provides beyond several other reports including their own that identify drug resistant Mpro mutations.

Reviewer #3: The manuscript titled “A comprehensive study of SARS-CoV-2 main protease (Mpro) inhibitor-resistant

mutants selected in a VSV-based system” by Costacurta et al presents new information on potential Nirmatrelvir resistant mutants using a VSV virus SARS-CoV-2 Mpro inhibitor-resistant mutant were selected using, VSV-Mpro, was applied to produce resistant mutants of nirmatrelvir and ensitrelvir safely and productively. Two live-cell-based protease mutation characterization assays were introduced to quantify the resistance phenotype of mutants. As expected, residues in catalytic site showed up infrequently, and the most frequent mutants were observed in positions further away from the active site. Some of the interesting mutants were thoroughly characterized, and results provide detailed results to better understand the mechanism of the interaction between the protease and inhibitors.

This prediction of resistant mutants of current FDA approved drug, such as nirmatrelvir (Paxlovid), will likely be important for new drug discovery. However, due to the limitations of VSV-Mpro system, the resistant mutants need to be confirmed by biochemical assay and even cell-based assay with authentic SARS-CoV2 mutant. However, this work goes beyond the scope of this paper. This paper is well written and should be accepted.

**Part II – Major Issues: Key Experiments Required for Acceptance**

Reviewer #1: Overall, the paper addresses an important topic as the potential emergence of drug resistance threatens our current antiviral therapies. The system used to identify mutations is clever, but it was described in a previous study. Thus, the impact of this paper is, for the most part, limited to the impact of the mutations identified in the serial passaging experiments. I have one major concern with this part of the study; the NSP5 variants identified were not tested in the context of authentic CoV replication. While I understand the reluctance to generate drug resistant mutations, especially in the omicron genetic background, these types of studies can be done in other ways (such as a non-infectious RNA replicon system). This type of analysis is critical, as the true importance of many of the identified mutants for the field currently remain unclear because the VSV system clearly selects for variants that are specific for that assay system.

Reviewer #2: 1. The reliance on theoretical models for insights into mutant vs wild-type Mpro structures, without performing well-established assays to measure enzyme kinetics, significantly weakens the impact of the manuscript. The modeling results are inadequate replacements for empirical data that can provide definitive quantitative measures of Ki, Km, kcat, and catalytic efficiencies. Conducting these measurements for the wild-type and mutants of Mpro (e.g. the mutants they analyzed by computational methods but also the key Omicron mutants), would lend credibility to their models. Specifically, measuring Km’s for model substrates, assessing the catalytic efficiency of the mutants relative to the wild-type, and Ki’s for nirmatrelvir and ensitrelvir will offer important insights into the mechanisms of resistance and structure-function relationships. For example, such data will shed light on the differences in substrate binding and inhibitor resistance, providing more clarity on why mutations such as L167F/P168S/L57F confer greater resistance to ensitrelvir compared to nirmatrelvir and how Omicron Mpro derived mutants differ from the Wuhan in its ability to maintain much of the normal proteolytic activity (inferred from their replication assays showing that Omicron derived mutants maintain nearly the same TM50 as the parental strain – Fig.3D). Although in the “Discussion” the authors suggest that individual mutations like L167F, P168S, S144A, and P184S could alter the binding pocket and reduce electrostatic interactions with nirmatrelvir, the specific effects on ensitrelvir binding is not discussed. Empirical enzymatic analyses will be crucial for understanding these interactions. The authors already show they are able to express and purify the respective mutants of Mpro so preforming the enzymatic assays is highly feasible.

2. The authors do not address the distinction between secondary and tertiary mutations that emerge in the Omicron Mpro with the P132H mutation versus those in the Wuhan Mpro. Furthermore, the authors have not explored the mechanistic underpinnings that could account for the observed differences in mutation-induced resistance to protease inhibitors. Elucidating how initial mutations can restrict the evolutionary trajectory of Mpro is of substantial interest, offering potential strategies to anticipate and counteract drug resistance. Applying the authors’ modeling approach to address this question would enhance the impact of this work. Performing enzymatic assays as suggested in (1) above will also provide needed insight. At a minimum, an analysis of mutants that are unique and only found to occur with the initial variant of Mpro versus those that are found in common would go a long way and perhaps shed important insight into the structural constraints of Mpro.

3. The computational models should include predictions for natural substrates (e.g. peptides that represent NSP4-5 junction) of Mpro. The authors can apply their modeling studies of resistant mutants to determine how binding to substrate might be altered compared to the WT or the initial mutant (e.g. L167F vs Omicron/207). Such information for example is important in their discussion (page 26, lines 5-14) describing the correlation between stability and replication rate – you can have a stable Mpro that nonetheless may have low proteolytic activity and therefore result in slower replication – substrate binding stability will be a better measure to correlate to the TM50’s.

Reviewer #3: (No Response)

**Part III – Minor Issues: Editorial and Data Presentation Modifications**

Reviewer #1: In figures 3 and 4, no statistical tests are performed which make it difficult to understand which changes are likely to be meaningful.

Reviewer #2: 1. 10-100uM of nirmatrelvir seems to be very high concentrations of drug to use in their experiments. Is expression of Mpro high in their cellular system? Is the difference due to the lack of using efflux inhibitors? Please comment on this in the discussion and in Methods. Could this also contribute to not picking up the T21I mutant?

2. It would be helpful in figure 1 or 2 to include a diagram indicating the dose of drug used for at each passage and the total passages. In Fig. 2B, indicate at what passage number and concentration of nirmatrelvir these mutants were obtained.

3. The mutants with >500 entries would be more informative if the authors included which CoV-2 variant they might be associated with (e.g. alpha, delta, omicron, etc) – if that information is available.

4. Nirmatrelvir and ensitrelvir inhibit Mpro by distinct mechanisms (as reported in Ref 42 – Noske et al. JBC 2023). Ensitrelvir is a nonpeptide noncovalent inhibitor and nirmatrelvir is a covalent peptidomimetic inhibitor. There should be some discussion about the different mechanisms of inhibition. For example, how might the mechanism of inhibition explain the difference in resistance to either drug displayed by L167F/P168S/L57F (Fig. 4D)?

Reviewer #3: (No Response)

PLOS authors have the option to publish the peer review history of their article (what does this mean?). If published, this will include your full peer review and any attached files.

Reviewer #1: No

Reviewer #2: No

Reviewer #3: No
---

## [Decision Letter · Decision Letter 1]

19 Aug 2024

Dear Heilmann,

We are pleased to inform you that your manuscript 'A comprehensive study of SARS-CoV-2 main protease (Mpro) inhibitor-resistant mutants selected in a VSV-based system' has been provisionally accepted for publication in PLOS Pathogens.

Best regards,

Mark T. Heise

Academic Editor

PLOS Pathogens

Meike Dittmann

Section Editor

PLOS Pathogens

Michael Malim

Editor-in-Chief

PLOS Pathogens

orcid.org/0000-0002-7699-2064

Reviewer Comments (if any, and for reference):

Reviewer's Responses to Questions

**Part I - Summary**

Reviewer #1: My pervious concerns have been addressed, I have no further comments.

Reviewer #2: I am satisfied with the authors' responses.

Reviewer #3: The authors have responded satisfactorily to the concerns of the reviewers and the paper is suitable for publication in PLOS Pathogens.

**Part II – Major Issues: Key Experiments Required for Acceptance**

Reviewer #1: (No Response)

Reviewer #2: None

Reviewer #3: (No Response)

**Part III – Minor Issues: Editorial and Data Presentation Modifications**

Reviewer #1: (No Response)

Reviewer #2: None

Reviewer #3: (No Response)

PLOS authors have the option to publish the peer review history of their article (what does this mean?). If published, this will include your full peer review and any attached files.

Reviewer #1: No

Reviewer #2: No

Reviewer #3: No

---

## [Editor Report · Acceptance letter]

3 Sep 2024

Dear Heilmann,

We are delighted to inform you that your manuscript, "A comprehensive study of SARS-CoV-2 main protease (Mpro) inhibitor-resistant mutants selected in a VSV-based system," has been formally accepted for publication in PLOS Pathogens.

Best regards,

Michael Malim

Editor-in-Chief

PLOS Pathogens

orcid.org/0000-0002-7699-2064